# CC-LEARN: COHORT-BASED CONSISTENCY LEARNING

## ABSTRACT

Large language models excel at many tasks but still struggle with consistent, robust reasoning. We introduce Cohort-based Consistency Learning (CC-LEARN), a reinforcement learning framework that trains on *cohorts of similar questions* instantiated from symbolic programmatic abstractions and executes a programmatic solution unchanged across each cohort. Our composite objective mixes execution-based signals with critique-based signals. The execution-based signals include cohort-level accuracy, retrieval usage, and penalties for invalid lookups. The critique-based signals come from a frozen judge that checks whether the program's sub-questions cover the key factors and whether its reasoning logic moves closer to a higher-quality self-improvement. Optimized via reinforcement learning, this objective steers the policy toward uniform, generalizable procedures rather than instance-specific shortcuts. Across five in-domain benchmarks (ARC-Easy/Challenge, CSQA, StrategyQA, HotpotQA) and three out-of-domain benchmarks (OpenBookQA, PubMedQA, MMLU), at two model scales (3B/7B), CC-LEARN delivers roughly 10–20 absolute-point gains over strong baselines under both lenient and strict criteria, improving accuracy and stabilizing reasoning. These results show that cohort-level RL with execution signals and external feedback effectively enforces cross-variant consistency in LLMs.

## 1 INTRODUCTION[1]

Large language models (LLMs) have made remarkable progress in complex reasoning tasks through strategies like prompting and step-by-step solution traces. Techniques such as *chain-of-thought* prompting (Wei et al., 2022) enable models to decompose problems into intermediate steps, significantly improving performance on arithmetic, commonsense, and various reasoning challenges. Similarly, decoding strategies like self-consistency (Wang et al., 2023) enhance accuracy by sampling multiple reasoning paths and selecting the most consistent answer across benchmarks. Despite these advances, LLMs frequently exhibit **inconsistency**: a model may correctly answer a question in one formulation but fail on a paraphrase or logically equivalent variant (Yu et al., 2024; Zhou et al., 2024; Li et al., 2024b). Moreover, even with the same answer, the underlying chain of reasoning can differ across variants. This behavior suggests brittle reasoning processes and undermines reliability in practical applications (McCoy et al., 2019; Geirhos et al., 2020).

Figure 1 illustrates this phenomenon with an example from StrategyQA. The original question asks:

> "Can you order an *Alfa Romeo* at *Starbucks*?"

A model may answer *No* by invoking a product-availability path (coffee shops do not sell cars / not on the menu). For a similar question,

> "Can you order a *Tesla* at *Dunkin' Donuts*?"

the same model may again answer *No* but justify it via a different partnership path (no business relationship ⇒ not available). Although both answers are correct, the model arrives at them via disjoint, partial reasoning paths and neither covering all possible factors. To solve cohorts of similar questions reliably and achieve genuine generalization, the model should enumerate the plausible conditions and integrate them into one single, reusable reasoning path, so correctness does not hinge

---

[1]Code and data will be released with the camera-ready version.

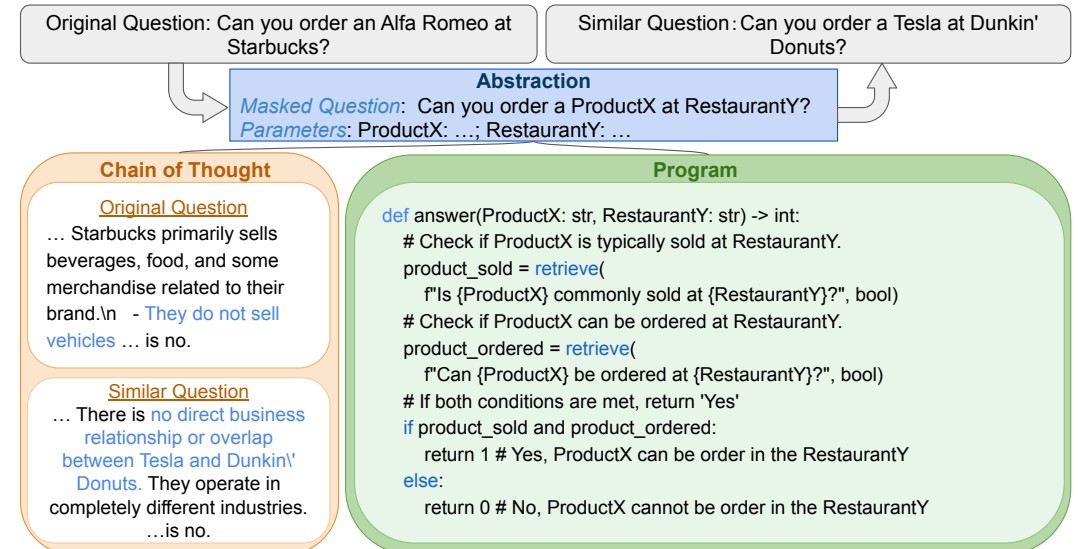

Figure 1: An illustrative example of cohort-based consistency learning(See Appendix A.3). (Top) The original question and a surface-variant question sharing the same reasoning path. (Middle) The masked abstraction template with its parameter dictionary, used to generate a cohort of factual variants. (Bottom) The executable program synthesized by the model, which issues simple `retrieve` calls for each substep and computes the answer, enforcing consistent reasoning across all cohort members.

on whichever partial test happens to fire (Ahn et al., 2025; McCoy et al., 2019; Geirhos et al., 2020). Such divergence across similar questions thus exemplifies reasoning inconsistency and underscores the need for training methods that explicitly enforce consistency across similar questions (Sinha et al., 2021; Zhao et al., 2021). Our goal, therefore, is to learn a unified, generalizable reasoning procedure rather than simply memorizing instance-level answers. Free-form chains of thought keep control flow implicit and tend to drift across variants, making it difficult to align and thus reward same reasoning steps across cohorts of similar questions. We therefore take inspiration from computer programs: they make control flow explicit, decompose reasoning into modular, reusable steps, and can execute the same reasoning steps across a cohort of similar questions (Gao et al., 2022; Chen et al., 2023; Yao et al., 2023). By granting reward only when the same procedure succeeds on most variants, consistency becomes the primary learning signal and shortcut solutions are disincentivized (Geirhos et al., 2020; McCoy et al., 2019).

Following such intuitions, we propose CC-LEARN, which trains LLMs on *cohorts* of similar questions expressed as executable programs, and optimizes a *cohort-level* reward via reinforcement learning. First, each question is transformed into a masked abstraction exposing its core reasoning structure (e.g., "Can you order a `ProductX` at `RestaurantY`?"). From this abstraction we automatically generate a cohort of factual variants: by requiring a single program to succeed on most or all variants during RL training, we eliminate cases where an incorrect reasoning path nonetheless produces the right answer by accident, directly enforcing true consistency. We then prompt the model to emit a compact program that issues only simple, atomic `retrieve` calls for each substep and uses a rejection filter to block any multi-step or invalid queries. This design uses the same simple `retrieve` calls for all cohort members, so the program runs unchanged on each variant; any change in output comes only from different facts, not from different reasoning steps. Finally, we apply Group Relative Policy Optimization (GRPO) (Shao et al., 2024) to maximize a composite signal that mixes execution-based signal with evaluator feedback from a frozen judge. The frozen judge rewards factorized coverage, discourages shortcuts, and steers learning to an improved program that works across the cohort and generalizes. This pipeline compels the model to learn uniform, verifiable reasoning procedures rather than exploiting shortcuts on individual instances.

Across five benchmarks—ARC-Easy, ARC-Challenge (Clark et al., 2018), StrategyQA (Geva et al., 2021), HotpotQA (Yang et al., 2018), and CommonsenseQA (Talmor et al., 2019)—CC-LEARN consistently outperforms SFT and RL baselines under both lenient and strict evaluation. On the 7B model, CC-LEARN improves over the strongest SFT baseline by roughly 20–35 absolute points on

ARC-Easy, CSQA, StrategyQA, and ARC-Challenge under both lenient and strict criteria. On the 3B model, CC-LEARN improves by roughly 20–31 points on ARC-Easy, ARC-Challenge, CSQA, and StrategyQA for both criteria. We further probe out-of-domain accuracy on three benchmarks-OpenBookQA (Mihaylov et al., 2018), PubMedQA (Jin et al., 2019), and MMLU (Hendrycks et al., 2021a;b)-CC-LEARN outperforms SFT and RL baselines by roughly 10-20 points. These trends, together with ablations on cohort-gated accuracy and disciplined retrieval and a small human preference study, indicate that training a single executable program across cohorts with judge critique yields more stable and consistent reasoning.

## 2 RELATED WORK

**Reasoning Consistency** LLMs often exhibit inconsistent reasoning when faced with paraphrased inputs. For example, prompts with similar surface familiarity but different underlying complexity yield divergent performance (Li et al., 2024a), and models may exploit spurious semantic cues rather than following intended chains of reasoning (Li et al., 2024b). To mitigate inconsistency, researchers have added training regularizers or auxiliary losses for paraphrase-invariance (Elazar et al., 2021; Zhou et al., 2022), leveraged knowledge graphs to generate paired questions for fine-tuning (Rajan et al., 2024), and applied self-consistency decoding to vote out illogical paths (Wang et al., 2023; Wei et al., 2022). Recent directions construct paraphrase/symmetry cohorts either at inference time (Chen et al., 2024)or for fine-tuning (Yao et al., 2025) to enforce that semantically equivalent inputs yield consistent outputs (Raj et al., 2025). Our work enforces consistent, programmatic reasoning across cohorts of similar questions that share the same reasoning path.

**Programmatic Abstractions for Reasoning** Programmatic or symbolic abstractions introduce a formal structure that can be executed and verified, improving transparency and reliability (Chen et al., 2024). Prior work translates questions into executable programs for verifiable stepwise reasoning (Gao et al., 2022; Chen et al., 2023), and ReAct interleaves reasoning with tool use to ground intermediate steps (Yao et al., 2023). Zhou et al. (2024; 2025) pursue conceptual/symbolic formulations, while Hong et al. (2024) requires an Abstraction-of-Thought plan before refining a concrete solution. Our approach similarly emits compact executable programs but differs by executing a *single* program unchanged across a cohort of similar questions.

**Reinforcement Learning for Enhanced Reasoning.** Reinforcement learning objectives can substantially boost an LLM's ability to solve multi-step problems by optimizing the reasoning path (Shen et al., 2025b; Xu et al., 2025b). Verifiable intermediate rewards catch and correct logical mistakes, leading to more stable reasoning (Xu et al., 2025a). Composite reward functions that blend answer accuracy, factuality, and faithfulness yield more dependable outputs (Wang et al., 2024), and potential-based shaping adds domain priors as soft constraints, speeding up training while keeping policies optimal (Nguyen et al., 2020). External verifiers and frozen judges have improved reliability by critiquing or selecting among candidate solutions (Cobbe et al., 2021), and recent RL systems explicitly train models to internalize search-like, stepwise procedures (Shen et al., 2025a). Together, these advances in RL-driven reasoning path optimization, verifiable reward design, and structured shaping inspire our cohort-level consistency framework.

## 3 METHOD

As shown in Figure 2, we first convert each question into a masked abstraction and instantiate a cohort of similar questions that share the same reasoning path (see Sec. 3.1 and Sec. 3.2). The policy is trained to emit one executable program that runs unchanged across the entire cohort; the program's only external operation is an atomic `retrieve(q, type)` call, which forces the reasoning path into code and keeps the set of `retrieve` calls invariant across variants. To prevent degenerate shortcuts, the retriever is fronted by a rejection prompt—ill-formed or non-atomic queries yield "idk" and incur a penalty. We optimize with GRPO on a composite signal mixing execution reward and critique reward using a simple three-role setup (policy, retriever, judge) (See Sec. 3.3). At test time, each synthesized program is executed over its cohort and scored under lenient ($\geq 4/6$) and strict ($\geq 5/6$) criteria (Sec. 3.4). Finally, Sec. 3.5 provides a simple analysis explaining why cohort-level rewards align with the consistency objective.

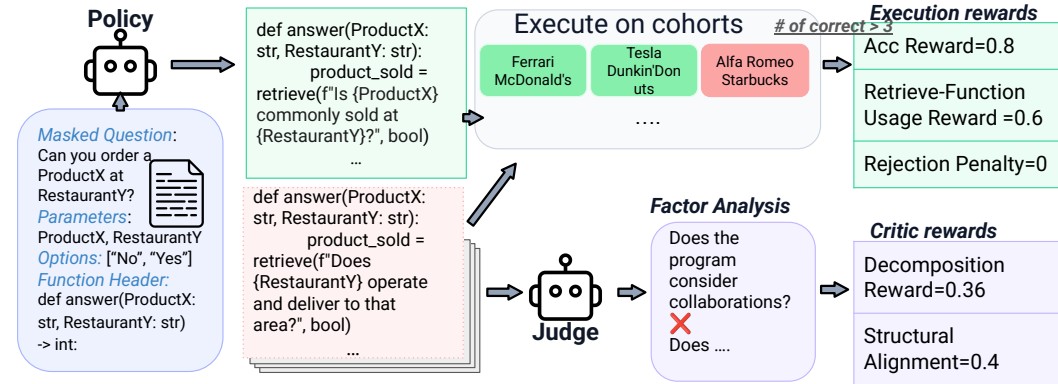

Figure 2: Overview of the CC-Learn training. For each masked abstraction, the policy emits a single program $p$ that uses only atomic `retrieve(q, type)` and is executed *unchanged* across a cohort of questions. Execution over the cohort yields the Accuracy Reward, the Retrieve-Function Usage Reward and Rejection Penalty. A judge supplies critique signals: the Factor-Complete Decomposition Reward and the Structural Alignment Reward. These combine into the composite RL objective.

## 3.1 PROGRAM GENERATION

We encode each reasoning path as a small Python function `def answer(param1: Type1, ..., paramK: TypeK)->int` whose body may use only atomic `retrieve(question: str, type)` calls and control flow. The model is given (i) a masked abstraction preserving logical structure, (ii) parameter names, (iii) the answer options, and (iv) the exact function header. The fixed header specifies parameter names and types as well as the return type, which (i) tells the model the concrete type of each input, guiding clearer program generation; and (ii) ensures unambiguous execution at evaluation time (we can call the same function signature on all variants). Four few-shot exemplars—boolean checks, numeric comparisons, list loops, and dependent lookups—specify the output format across cohort abstractions (Appendix A.1.5). At generation time the policy fills in the function body; at execution time each `retrieve` is issued to the retriever with rejection filtering.

## 3.2 DATA GENERATION AND PREPROCESSING

To foster generalizable reasoning and provide a strong foundation for our models, we employ a high-quality data preparation pipeline.

**Similar Questions Generation** Central to our approach is the construction of cohorts of similar questions that share same reasoning paths but differ in factual content. This process begins with a corpus of 5,000 original questions randomly sampled from each domain's training split (1,000 per domain). As Figure 1 shows, for each original question, we generate an abstraction by creating an abstraction that preserves its core reasoning structure while parameterizing entities, allowing for the substitution of different facts (see Appendix A.1.2). Using LLaMA-3.3-70B-Instruct (Team, 2024a), we then instantiate 5 similar questions for each abstraction(see Appendix A.1.3). These variants are solvable through the same reasoning path dictated by the abstraction but feature different specific entities and details. To ensure the fidelity of our training data, answers to all generated similar questions are cross-validated using three state-of-the-art models: LLaMA-3.3-70B-Instruct, DeepSeek-R1-Distill-Llama-70B(DeepSeek-AI, 2025), and Qwen-2.5-72B-Instruct(Yang et al., 2024; Team, 2024b). This process is supplemented by human verification of both the validity of similar questions and the correctness of their labels, as detailed in 4.5.

**Program Corpus for Supervised Fine-Tuning (SFT).** We construct two corpora corresponding to the SFT variants. For SFT$_{HQ-500}$, we synthesize 500 programs (100 per domain) with LLaMA-3.3-70B-Instruct and rigorously verify each to achieve **0% rejection** and **100% execution accuracy** on its question. For SFT$_{DM-5k}$, we reuse the 5,000 RL-training instances and generate programs with

the same 70B model and prompts; to keep computational cost tractable at this scale, we retain basic validity checks but do not perform the exhaustive curation needed to guarantee 0% rejection.

### 3.3 COHORT-BASED REINFORCEMENT LEARNING

We optimize the policy with Group Relative Policy Optimization (GRPO) on cohorts of six (1 original + 5 similar). For each abstraction, the policy emits a single executable program $p$ that is executed *unchanged* across all variants. A retriever answers atomic `retrieve` calls, and a rejection filter blocks multi-hop or invalid queries. The policy then receives a composite reward $R$. We group the rewards into two families: (i) *execution-based* rewards (Accuracy Reward, the Retrieve-Function Usage Reward and Rejection Penalty), which are verifiable from program execution and the retriever/rejection outcomes; and (ii) *critique-based* rewards (the Factor-Complete Decomposition Reward and the Structural Alignment Reward), which are produced by a frozen judge model. Together, these rewards guide the policy toward accurate, disciplined decomposition and convergence to factor-complete reasoning procedures across the cohort.

**Model Architecture** We evaluate both 3B and 7B model sizes throughout. Our framework employs three complementary language models: **(a) Policy Model:** Qwen-2.5-Coder-Instruct (3B/7B) (Yang et al., 2024; Team, 2024b; Hui et al., 2024) generates structured, executable programs for abstracted questions, specifying the reasoning path. **(b) Retriever Model:** Qwen-2.5-Instruct (3B/7B) serves as our retriever model; it executes the simple retrieval calls generated within the programs by the policy model. **(c) Judge Model:** the same Qwen-2.5-Instruct (3B/7B) checkpoint as the retriever, used as an evaluator to score whether sub-questions cover key factors and to propose a concise improved program $p^+$ for logic-level alignment checking. This architectural separation is identical across scales and ensures that the policy must formulate its reasoning strategy without direct access to factual information during program generation.

**Rejection Prompts in Retrieval** To encourage the policy model to learn robust, generalizable reasoning and prevent the policy model from circumventing the intended reasoning process by issuing trivial or multi-step queries—behavior akin to the "deceptive shortcuts" observed in prior work (Li et al., 2024b)—we equip the retriever model with a rejection-prompt filter. With a few-shot prompt (see Appendix A.1.1), the retriever only accepts straightforward, single-step factual questions (e.g., Is {ProductX} sold at {RestaurantY}?) and replies with "idk" to any multi-step or invalid queries. Any rejected call will incur the penalty $R_{\text{rej}}$, thereby incentivizing the policy model to offload only elementary lookups and to internalize the full reasoning chain within the generated program.

**Interactive Training Pipeline.** The GRPO training loop proceeds interactively. First, the policy model proposes a program in response to an abstracted question. Second, this program is executed on both the original question and its five similar variants. Any information retrieval calls embedded in the program are handled by the retriever model. Finally, based on the execution outcomes across the entire question family, a scalar reward is computed. This reward considers accuracy, retrieval efficiency, and the rejection ratios.

**Reward Components.** Our composite reward function $R$ is designed to guide the model toward effective and generalizable reasoning. It is defined as $R = R_{\text{acc}} + R_{\text{ret}} + R_{\text{rej}} + R_{\text{fc}} + R_{\text{sa}}$. and comprises two families: *execution-based* signals $\left(R_{\text{acc}}, R_{\text{ret}}, R_{\text{rej}}\right)$ that are directly verifiable from program execution, and *critique-based* signals $\left(R_{\text{fc}}, R_{\text{sa}}\right)$ that come from a frozen judge model.

- **Accuracy Reward ($R_{\text{acc}}$):** This reward is calculated as $R_{\text{acc}} = 0.2 \cdot n_{\text{correct}}$, where $n_{\text{correct}} \in \{0, 1, \ldots, 6\}$ is the number of correctly answered questions within the cohort of similar questions (1 original + 5 similar). This yields $R_{\text{acc}} \in [0, 1.2]$. We scale this term to 0.2 per correct so that gaining a few additional correct variants outweighs any single term, thereby aligning training with the cohort-consistency objective rather than stylistic surrogates.

- **Retrieve-Function Usage Reward ($R_{\text{ret}}$):** This component encourages appropriate problem decomposition and is assigned based on the number of retrieval calls ($n_{\text{calls}}$) made by the

program:

$$R_{\text{ret}} = \begin{cases} -0.6 & \text{if } n_{\text{calls}} = 0 \\ 0 & \text{if } n_{\text{calls}} = 1 \\ +0.6 & \text{if } n_{\text{calls}} > 1 \end{cases}$$

This results in $R_{\text{ret}} \in \{-0.6, 0, 0.6\}$. The retrieve-function usage reward encourages problem decomposition and discourages trivial solutions with no retrieval calls.

- **Rejection Penalty** ($R_{\text{rej}}$): This penalty discourages ineffective retrieval calls and is given by $R_{\text{rej}} = -0.1 \cdot n_{\text{rejected}}$, where $n_{\text{rejected}} \in \{0, 1, \ldots, 6\}$ is the number of questions in the group whose retrieve call is rejected by the retrieve model. This leads to $R_{\text{rej}} \in [-0.6, 0]$. The Rejection Penalty penalizes attempts to re-ask the original question or formulate overly similar questions, forcing multi-step, valid reasoning.

- **Factor-Complete Decomposition Reward**($R_{\text{fc}}$): Given the program, the full set of similar questions, and failure patterns, a judging model scores whether the program's `retrieve` sub-questions cover the key factors needed to solve the original abstraction. Let $s_{\text{fc}} \in [1, 10]$ be this judged coverage score; we map it to $R_{\text{fc}} = 0.06 \cdot s_{\text{fc}}$, which yields $R_{\text{fc}} \in [0.06, 0.6]$. This term rewards factor-complete decompositions even when some variants remain wrong.

- **Structural Alignment Reward**($R_{\text{sa}}$): Given the full context (cohort, original program $p$, failure patterns), a judge model proposes an improved program $p^+$. We then compare $p$ and $p^+$ at the *logic level*: after stripping comments and formatting, we compute an AST-level structural similarity $s_{\text{logic}} \in [0, 1]$. We set $R_{\text{sa}} = 0.6 \cdot s_{\text{logic}} \in [0, 0.6]$.

  By construction, higher $R_{\text{sa}}$ indicates that, after reflection, the logic of the original program $p$ is becoming *closer* to that of its improved counterpart $p^+$—i.e., alignment in logiv increases over training—rather than merely matching on superficial patterns.

**RL Variants** To isolate how the accuracy signal shapes learning, we consider two training-time variants that differ only in how $R_{\text{acc}}$ is computed (all other terms unchanged; to avoid confusion, *lenient/strict* refer only to evaluation criteria): **(a) Cohort Accuracy:** Accuracy-based rewards ($R_{\text{acc}}$) are granted only if the generated program successfully answers at least 4 out of the 6 questions in one group. This enforces a higher standard of generalizability. **(b) Normal Accuracy:** Accuracy rewards ($R_{\text{acc}}$) are granted for every successful program execution on a question within the group, regardless of performance on other questions in that group. This provides a more granular learning signal.

### 3.4 COHORT EXECUTION TEST

We evaluate each program on a cohort of six questions (original + five variants). We report two criteria: **Strict Accuracy**—the program is correct iff it answers $\geq 5$ of 6; **Lenient Accuracy**—correct iff it answers $\geq 4$ of 6. In addition, we apply a lightweight rejection check at evaluation to prevent degenerate behavior that simply re-asks the original question or violates the simple-fact constraint; any instance with a rejected `retrieve` call is counted as incorrect under both criteria.

### 3.5 WHY COHORTS HELP: A SIMPLE THEORETICAL ANALYSIS

**Setup.** Fix a cohort of $N$ variants $\{x_i\}_{i=1}^{N}$ from an abstraction. A program $p$ is executed *unchanged* on all variants. Let $Z_i(p) \in \{0, 1\}$ indicate correctness on $x_i$ (with the retriever and rejection filter). Define $S(p) = \sum_{i=1}^{N} Z_i(p)$ and the evaluation metric

$$J_K(p) := \Pr\big[S(p) \geq K\big],$$

where $K = 4$ (lenient) or $5$ (strict) in our experiments.

We compare two training surrogates: **1)** Normal accuracy: $R_{\text{normal}}(p) = \mathbb{E}\big[\frac{1}{N} \sum_{i=1}^{N} Z_i(p)\big]$; **2)** Cohort accuracy: $R_{\text{cohort}}(p) = \mathbb{E}\big[\mathbf{1}\{S(p) \geq K\}\big]$.

**Proposition 1 (Exact alignment).** $R_{\text{cohort}}(p) = J_K(p)$ for every $p$. Hence maximizing the cohort reward exactly maximizes the $K$-of-$N$ consistency objective used at evaluation.

*Proof.* Immediate from the definition. $\square$

**Proposition 2 (Normal accuracy is an inconsistent surrogate for $K$-of-$N$).** There exist programs $p, q$ such that $R_{\text{normal}}(q) > R_{\text{normal}}(p)$ but $J_K(q) < J_K(p)$.

*Intuition.* Raising the mean per-variant success by trading one "hard" variant for several "easier" ones can lower the chance that enough variants succeed simultaneously. A concrete counterexample and numbers are in App. A.4.

## 4 EXPERIMENTS

In this section, we provide detailed experimental settings and results that highlight the effectiveness of our RL framework in training LLMs to perform transparent reasoning through structured, executable programs. The full set of hyperparameters is listed in Appendix A.2.

**Baselines.** We compare our cohort-based RL models against two configurations: **(a) Vanilla Model:** the off-the-shelf Qwen-2.5-Coder-Instruct checkpoint, used without any additional supervised fine-tuning or RL. **(b) Supervised Fine-Tuning (SFT):** we consider two SFT variants on the same backbone. $\text{SFT}_{\text{HQ-500}}$ ("High Quality SFT") fine-tunes on a curated set of 500 exemplar programs (100 per domain) until held-out loss stabilizes (typically ~1.1 epochs; see Sec. 3.2). $\text{SFT}_{\text{DM-5k}}$ ("Data-Matched SFT") fine-tunes on the same 5,000 instances used for RL, using programs synthesized by LLaMA-3.3-70B-Instruct with the same prompting but without exhaustive curation, isolating the effect of RL's cohort-level credit assignment from simply scaling SFT on the RL data distribution.

### 4.1 EXPERIMENTAL SETUP

**Datasets and Test Set** We evaluate on five publicly available benchmarks: ARC-Easy, ARC-Challenge, CSQA, StrategyQA, and HotpotQA. Our test set comprises random 2,500 questions in total (500 per dataset, randomly sampled from dev/test split), each paired with five similar questions generated by our abstraction pipeline.

**Evaluation Protocol** For each of the 2,500 test questions, we draw 11 samples from the policy, execute each sampled program on its six-question cohort, and aggregate predictions via self-consistency: the final answer is the majority vote over the 11 runs. We then compute strict and lenient accuracy as defined in Sec. 3.4. This protocol enables us to report not only standard accuracy but also the critical generalization-across-variants metrics.

### 4.2 MAIN RESULTS

On 7B, $\text{RL}_{\text{Cohort}}$ (Exec+Crit) improves over the strongest SFT baseline ($\text{SFT}_{\text{DM-5k}}$) by +33.2 on ARC-Easy (74.8 vs. 41.6), +37.4 on CSQA (73.4 vs. 36.0), and +28.2 on StrategyQA (45.8 vs. 17.6) under the lenient metric; Under the strict metric, gains remain large: +36.8 (ARC-Easy: 68.0 vs. 31.2), +35.2 (CSQA: 62.4 vs. 27.2), +22.4 (StrategyQA: 31.0 vs. 8.6), and +36.2 (ARC-Challenge: 56.6 vs. 20.4). HotpotQA and ARC-Challenge are exceptions: under the lenient metric, $\text{RL}_{\text{Cohort}}$ (Exec+Crit) is slightly below the strongest RL variant. Under the strict metric, HotpotQA still trails ($-5.2$: 54.2 vs. 59.4 with execution-only), while ARC-Challenge is effectively tied (56.6 vs. 56.2).

On 3B, $\text{RL}_{\text{Cohort}}$ (Exec+Crit) improves over $\text{SFT}_{\text{DM-5k}}$ by +20.2 (ARC-Easy: 40.8 vs. 20.6), +28.6 (ARC-Challenge: 41.8 vs. 13.2), and +15.0 (CSQA: 42.6 vs. 27.6)under the lenient metric; strict gains are +23.0 (ARC-Easy), +26.6 (ARC-Challenge), +17.4 (CSQA), and +31.0 (StrategyQA). HotpotQA shows little change at 3B ($-3.2$ lenient; +0.8 strict). Taken together (Tables 1 and 2), these gains are consistent across criteria and model sizes, indicating that enforcing a single executable program with critique yields more stable, consistent reasoning on complex, multi-step questions.

### 4.3 SANITY CHECK EXPERIMENTS

**Reject Prompts Analysis** We validate the retriever's rejection prompts by measuring rejection ratios on 5k training questions and on SimpleQA (Wei et al., 2024) to check over-rejection. As shown in Table 3, multi-step questions are rejected far more often than SimpleQA, indicating the mechanism efficiently distinguishes question types and curbs rephrasing-based retrieval shortcuts.

| Model | Methods | ARC-Challenge | ARC-Easy | CSQA | StrategyQA | HotpotQA |
|---|---|---|---|---|---|---|
| Qwen2.5-Coder-7B-Instruct | Vanilla | $19.0 \pm 3.1$ | $30.0 \pm 4.0$ | $29.8 \pm 4.0$ | $12.6 \pm 2.9$ | $45.0 \pm 4.3$ |
| | $\text{SFT}_{\text{HQ-500}}$ | $19.8 \pm 3.5$ | $33.4 \pm 4.1$ | $32.0 \pm 4.1$ | $12.0 \pm 2.9$ | $46.8 \pm 4.4$ |
| | $\text{SFT}_{\text{DM-5k}}$ | $28.8 \pm 4.0$ | $41.6 \pm 4.3$ | $36.0 \pm 4.2$ | $17.6 \pm 3.3$ | $77.6 \pm 3.7$ |
| | $\text{RL}_{\text{Normal}}$ (Acc) | $30.2 \pm 4.0$ | $43.4 \pm 4.3$ | $36.0 \pm 4.2$ | $15.8 \pm 3.2$ | $73.2 \pm 3.8$ |
| | $\text{RL}_{\text{Cohort}}$ (Exec) | $51.0 \pm 4.4$ | $60.8 \pm 4.3$ | $59.6 \pm 4.3$ | $38.2 \pm 4.2$ | $\mathbf{81.4 \pm 3.4}$ |
| | $\text{RL}_{\text{Normal}}$ (Exec+Crit) | $\mathbf{66.2 \pm 4.1}$ | $72.6 \pm 3.9$ | $68.0 \pm 4.1$ | $\underline{43.6 \pm 4.3}$ | $81.0 \pm 3.4$ |
| | $\text{RL}_{\text{Cohort}}$ (Exec+Crit) | $\underline{65.8 \pm 4.1}$ | $\mathbf{74.8 \pm 3.8}$ | $\mathbf{73.4 \pm 3.9}$ | $\mathbf{45.8 \pm 4.4}$ | $79.4 \pm 3.5$ |
| Qwen2.5-Coder-3B-Instruct | Vanilla | $12.2 \pm 2.9$ | $14.4 \pm 3.1$ | $12.0 \pm 2.9$ | $\underline{5.4 \pm 2.0}$ | $17.0 \pm 3.3$ |
| | $\text{SFT}_{\text{DM-5k}}$ | $13.2 \pm 3.0$ | $20.6 \pm 3.5$ | $27.6 \pm 3.9$ | $3.6 \pm 1.7$ | $\mathbf{18.2 \pm 3.4}$ |
| | $\text{RL}_{\text{Cohort}}$ (Exec) | $\underline{31.6 \pm 4.1}$ | $30.4 \pm 4.0$ | $38.0 \pm 4.2$ | $1.8 \pm 1.2$ | $0.4 \pm 0.7$ |
| | $\text{RL}_{\text{Cohort}}$ (Exec+Crit) | $\mathbf{41.8 \pm 4.3}$ | $\mathbf{40.8 \pm 4.3}$ | $\mathbf{42.6 \pm 4.3}$ | $\mathbf{44.2 \pm 4.3}$ | $15.0 \pm 3.1$ |

Table 1: Lenient Accuracy (%) across datasets. Vanilla: off-the-shelf Qwen-2.5-Coder-Instruct. $\text{SFT}_{\text{HQ-500}}$/$\text{SFT}_{\text{DM-5k}}$: curated 500 exemplars vs. data-matched 5k set. $\text{RL}_{\text{Normal}}$ (Acc): RL with accuracy-only reward ($R_{\text{acc}}$; baseline). $\text{RL}_{\text{Cohort}}$ (Exec): RL with execution-based rewards only ($R_{\text{acc}}, R_{\text{ret}}, R_{\text{rej}}$). $\text{RL}_{\text{Normal}}$/$\text{RL}_{\text{Cohort}}$(Exec+Crit): RL using both execution- and critique-based rewards ($R_{\text{fc}}, R_{\text{sa}}$), with per-instance vs. cohort-gated accuracy. Here $\pm$ denotes the half-width of the 95% Wilson confidence interval. **Bold** = best, underline = second best.

| Model | Methods | ARC-Challenge | ARC-Easy | CSQA | StrategyQA | HotpotQA |
|---|---|---|---|---|---|---|
| Qwen2.5-Coder-7B-Instruct | Vanilla | $13.6 \pm 2.9$ | $22.6 \pm 3.7$ | $20.6 \pm 3.5$ | $6.8 \pm 2.2$ | $27.2 \pm 3.9$ |
| | $\text{SFT}_{\text{HQ-500}}$ | $14.4 \pm 3.1$ | $24.2 \pm 3.7$ | $25.0 \pm 3.8$ | $6.2 \pm 2.1$ | $27.8 \pm 3.9$ |
| | $\text{SFT}_{\text{DM-5k}}$ | $20.4 \pm 3.5$ | $31.2 \pm 4.1$ | $27.2 \pm 3.9$ | $8.6 \pm 2.5$ | $53.2 \pm 4.3$ |
| | $\text{RL}_{\text{Normal}}$ (Acc) | $22.4 \pm 3.7$ | $35.4 \pm 4.2$ | $27.6 \pm 3.9$ | $8.4 \pm 2.4$ | $49.8 \pm 4.4$ |
| | $\text{RL}_{\text{Cohort}}$ (Exec) | $40.6 \pm 4.3$ | $51.2 \pm 4.4$ | $48.4 \pm 4.4$ | $22.2 \pm 3.7$ | $\mathbf{59.4 \pm 4.3}$ |
| | $\text{RL}_{\text{Normal}}$ (Exec+Crit) | $\underline{56.2 \pm 4.3}$ | $64.4 \pm 4.2$ | $59.8 \pm 4.3$ | $\underline{27.2 \pm 3.9}$ | $\underline{56.2 \pm 4.3}$ |
| | $\text{RL}_{\text{Cohort}}$ (Exec+Crit) | $\mathbf{56.6 \pm 4.3}$ | $\mathbf{68.0 \pm 4.1}$ | $\mathbf{62.4 \pm 4.2}$ | $\mathbf{31.0 \pm 4.0}$ | $54.2 \pm 4.4$ |
| Qwen2.5-Coder-3B-Instruct | Vanilla | $7.6 \pm 2.3$ | $9.8 \pm 2.6$ | $8.2 \pm 2.4$ | $1.2 \pm 1.0$ | $6.0 \pm 2.1$ |
| | $\text{SFT}_{\text{DM-5k}}$ | $8.8 \pm 2.5$ | $14.0 \pm 3.0$ | $18.8 \pm 3.4$ | $\underline{1.6 \pm 1.2}$ | $\underline{6.8 \pm 2.2}$ |
| | $\text{RL}_{\text{Cohort}}$ (Exec) | $\underline{26.4 \pm 3.9}$ | $\underline{26.6 \pm 3.9}$ | $\underline{32.4 \pm 4.1}$ | $1.4 \pm 1.1$ | $0.2 \pm 0.3$ |
| | $\text{RL}_{\text{Cohort}}$ (Exec+Crit) | $\mathbf{35.4 \pm 4.2}$ | $\mathbf{37.0 \pm 4.2}$ | $\mathbf{36.2 \pm 4.2}$ | $\mathbf{32.6 \pm 4.1}$ | $\mathbf{7.6 \pm 2.3}$ |

Table 2: Strict Accuracy (%) across datasets. Naming and notes follow Table 1. **Bold** = best, underline = second best.

**Upper Bound Analysis** To assess the theoretical feasibility of our evaluation protocol, we estimate a theoretical upper-bound performance by running pass@128 on a random subset of 50 questions per domain and then manually correcting any incorrect programs. As shown in Table 3, after this minor intervention the model reaches near-perfect accuracy. Crucially, this demonstrates that the evaluation itself is not an insurmountable "mission impossible"; rather, the large gap between these upper-bound scores and our current best results underscores that existing LLMs still fall well short of their potential and require significant advances in reasoning consistency and generalization.

## 4.4 ABLATION STUDIES

**Training on Original Questions Only.** To further evaluate the effectiveness of the similar question during training, we conducted an ablation study where RL training was performed solely on the original questions without considering the similar variants. Table 4 and Table 5 compare the performance of this approach with our Cohort RL variant. The results demonstrate consistent performance degradation when training solely on original questions, particularly for complex reasoning tasks like ARC-Challenge and CSQA. This confirms that the similar questions play a crucial role in compelling the model to learn generalizable reasoning strategies rather than question-specific shortcuts.

**High-Quality Retriever & Judge** To further probe how retriever/judge strength affects training, we run a controlled study where all models are trained at 3B. Concretely: 3B-3B trains with a 3B retriever+judge and evaluates with a 3B retriever; 3B-7B trains with a 3B retriever+judge but

evaluates with a 7B retriever; 7B-7B trains with a 7B retriever+judge and evaluates with a 7B retriever. As shown in Table 6 and Table 7, across different datasets under lenient/strict metrics, 7B-7B yields the strongest scores on most datasets, while 3B-7B is consistently second-best and improves over 3B-3B. An exception is StrategyQA, where lighter judges (3B-3B / 3B-7B) edge out 7B-7B. Overall, stronger retrieval/judging generally boosts reasoning, but dataset dynamics can favor smaller judges.

**Out Of Domain data Accuracy**    To assess generalization beyond our training domains, we evaluated on three out of domain benchmarks—OpenBookQA, PubMedQA, and MMLU (Table 8). Because constructing reliable similar-question cohorts is nontrivial for these datasets, we report single-question accuracy only, using the same self-consistency strategy as in our in-domain evaluation. $RL_{Cohort}$ (Exec+Crit) attains the best accuracy on PubMedQA and MMLU and is on par with $RL_{Normal}$ (Exec+Crit) on OpenBookQA; both RL variants substantially outperform $SFT_{DM-5k}$ and Vanilla across all three datasets. These results indicate that CC-LEARN not only stabilizes in-domain reasoning but also transfers effectively to new tasks without cohort construction.

### 4.5    HUMAN STUDY

**Similar Question Quality Assessment**    To verify that our automated pipeline produces high-quality question variants, we assigned five annotators to assess a random sample of 150 generated questions (30 per domain) along two criteria: whether the answer label matches the ground truth, and whether the question follows its abstraction. The detailed results are summarized in Table 9. As shown in Table 9, our generated similar questions have high label and abstraction accuracy($\geq 90\%$).

**Comparison of Reasoning Path**    We also conducted a comparison of programmatic reasoning paths from our RL-trained model versus a supervised fine-tuning (SFT) baseline. Two annotators evaluated 100 instances (20 per domain), choosing which program showed superior logical coherence, clarity, and decomposition. Results are included in Table 9. As shown in Table 9, the Cohort RL's reasoning paths are preferred 47% of the time over the left 2 models.

**Failure Case Analysis**    We analyze 50 shared failure questions(10 per domain) where both SFT and RL answered incorrectly, and label each with one of three mutually exclusive types: **Ambiguity/Annotation**: the question or the label is ambiguity; **Control-Flow/Syntax**: the program's explicit logic is wrong (e.g., AND/OR aggregation, quantifier handling, branching/looping) or syntax mishaps lead to an incorrect branch; **Retrieve**: misuse of the `retrieve` call (non-atomic or multi-hop queries, re-asking the original question, or queries that trigger rejection). Results are included in Table 10. Two annotators independently labeled all items. As shown in the Table 10, RL reduces retrieve failures, consistent with execution signals and rejection penalties discouraging non-atomic or shortcut queries. Residual RL errors shift toward Control-Flow/Syntax. This supports our claim that cohort-gated RL suppresses retrieval shortcuts and enforces more disciplined, programmatic reasoning.

## 5    CONCLUSION

We present a training framework that improves the consistency and reliability of LLM reasoning by forming cohorts of similar questions and training a single executable program to run unchanged across each cohort, optimized with a composite RL objective that combines execution feedback and judge critique. This directly addresses inconsistent answers across similar inputs by rewarding uniform, factor-complete procedures instead of instance-specific shortcuts. Ablation analyses indicate that cohort-gated accuracy better aligns training with the $K$-of-$N$ objective than instance-wise rewards, and the rejection-filtered retriever curbs shortcut queries, yielding more disciplined decompositions. Across five in-domain benchmarks and three out-of-domain benchmarks, the method outperforms vanilla, supervised fine-tuning and RL baselines under both lenient and strict evaluations, with clear gains on complex multi-step tasks; human evaluation further prefers our reasoning paths. Together, these results indicate that cohort-gated, program-based RL effectively instills generalizable, stable reasoning.

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

## A  APPENDIX

You may include other additional sections here.

### A.1  PROMPTS

#### A.1.1  REJECTION PROMPT

---

**Prompt for *Fact-Lookup Assistant***

**System:** You are a fact-lookup assistant. For each user query, first decide if it's a simple, single-step fact lookup without solving it and then return a JSON object with exactly one key, "answer", wrapped in "'json ...'". Match the type specified in parentheses (int, str, list, bool). If a query requires more than a straightforward fact check or true/false lookup—for example, multi-step reasoning or subjective judgment—reply with "idk".

**User:** Who finished immediately after the winner at the 1992 Olympic 100m final? (str)
**Assistant:** [Explanation] You must identify the winner, then determine who came second—this isn't single-step. "'json "answer": "idk" "'

**User:** How many planets are in the solar system? (int)
**Assistant:** [Explanation] Simple fact check. "'json "answer": 8 "'

**User:** What is the profession of Michael Jackson? (str)
**Assistant:** [Explanation] Single well-known profession of a public figure. "'json "answer": "singer" "'

**User:** Who has more than one Nobel Prize? (list)
**Assistant:** [Explanation] Factual list of individuals with multiple Nobel Prizes. "'json "answer": ["John Bardeen", "Frederick Sanger", "Linus Pauling", "Marie Curie"] "'

**User:** Is the CEO of Tesla older than the current President of France? (bool)
**Assistant:** [Explanation] Requires fetching and comparing two birthdates—multi-step. "'json "answer": "idk" "'

**User:** Is the Eiffel Tower located in Paris, France? (bool)
**Assistant:** [Explanation] Single-step landmark location. "'json "answer": true "'

**User:** Did England win any Olympic gold medals in 1800? (bool)
**Assistant:** [Explanation] Must check when the modern Olympics began and then medal records—multi-step. "'json "answer": "idk" "'

**User:** What is the population of the largest country entirely south of the equator? (int)
**Assistant:** [Explanation] Identify the country then lookup its population—multi-step. "'json "answer": "idk" "'

**User:** List the U.S. states admitted to the Union between the first and the last of the original 13 colonies. (list)
**Assistant:** [Explanation] Order states by admission date and filter—multi-step. "'json "answer": "idk" "'

**User:** Can food be cooked in the cosmic microwave background? (bool)
**Assistant:** [Explanation] Must compare CMB temperature ( 2.7 K) to cooking physics—multi-step. "'json "answer": "idk" "'

**User:** Are Waris Hussein and Mathieu Kassovitz both actors? (bool)
**Assistant:** [Explanation] Fetch each person's profession and compare—multi-step. "'json "answer": "idk" "'

**User:** {prompt} ({return_type})
**Assistant:**

---

### A.1.2 ABSTRACTION GENERATION PROMPT

---

**Prompt for** *Masked Question Generation*

**System:** Generate a masked question and corresponding parameters for the given question. These generations will be used to create similar questions that share the same reasoning path and options. The similar questions can only replace the values in the parameters. Therefore, be careful with the parameters. Do not include options or answers in the parameters. Present the output in JSON format.

**User:** The Sugarcubes and SCH, are related to music? Options: A. No B. Yes. Answer: B
**Assistant:**

```
{
  "masked_question": "Are Entity1 and Entity2 both related to DomainX?",
  "parameters": {
    "Entity1": "The Sugarcubes",
    "Entity2": "SCH",
    "DomainX": "music"
  }
}
```

**User:** Are Ken Kwapis and Jan Švankmajer both filmmakers? Options: A. No B. Yes. Answer: B
**Assistant:**

```
{
  "masked_question": "Are Person1 and Person2 both ProfessionX?",
  "parameters": {
    "Person1": "Ken Kwapis",
    "Person2": "Jan Švankmajer",
    "ProfessionX": "filmmakers"
  }
}
```

**User:** Which of these is one similarity of the elements in the first group of the periodic table? Options: A. The elements in the group are very unreactive metals. B. The elements in the group react with oxygen in similar ways. Answer: B
**Assistant:**

```
{
  "masked_question": "Which of these is one similarity of the elements in the Group1 of the periodic table?",
  "parameters": {
    "Group1": "first group"
  }
}
```

**User:** Which celestial object listed below has the greatest density? Options: A. a planet B. a neutron star. Answer: B
**Assistant:**

```
{
  "masked_question": "Which celestial object listed below has the Superlative Property?",
  "parameters": {
    "Superlative": "greatest",
    "Property": "density"
  }
}
```

**User:** On what would you serve an egg to someone? Options: A. plate B. chicken coop. Answer: A
**Assistant:**

```
{
  "masked_question": "On what would you serve a FoodItem to someone?",
  "parameters": {
    "FoodItem": "egg"
  }
}
```

**User:** Fighting inflation means being aware of many things, such as what with loans? Options: A. rising interest rates B. loss of money. Answer: A
**Assistant:**

```
{
  "masked_question": "Fighting EconomicIssue means being aware of many things, such as what with FinancialAspect?",
  "parameters": {
    "EconomicIssue": "inflation",
    "FinancialAspect": "loans"
  }
}
```

**User:** Question: {question} Options: {options}. Answer: {answer}
**Assistant:**

---

### A.1.3 SIMILAR QUESTION PROMPT

---

**Prompt for *Similar Question Generation***

**System:** Generate five new questions that follow the same underlying reasoning process based on the masked questions, parameters, and the original question. Ensure that the choices and masked question remain the same as the original. You may only change the values within the `parameters` dictionary. Present the output in JSON format.

**User:** The Sugarcubes and SCH, are related to music? Options: A. No B. Yes. Answer: B. Masked Question: Are Entity1 and Entity2 both related to DomainX? Parameters: 'Entity1': 'The Sugarcubes', 'Entity2': 'SCH', 'DomainX': 'music'.
**Assistant:**

```
{
  "similar_questions": [
    {
      "question": "Are The Beatles and ABBA both related to music?",
      "answer": "B",
      "parameters": {
        "Entity1": "The Beatles",
        "Entity2": "ABBA",
        "DomainX": "music"
      }
    },
    {
      "question": "Are Marvel and DC both related to comics?",
      "answer": "B",
      "parameters": {
        "Entity1": "Marvel",
        "Entity2": "DC",
        "DomainX": "comics"
      }
    },
    {
      "question": "Are NASA and SpaceX both related to space exploration?",
      "answer": "B",
      "parameters": {
        "Entity1": "NASA",
        "Entity2": "SpaceX",
        "DomainX": "space exploration"
      }
    },
    {
      "question": "Are Pepsi and Coca-Cola both related to beverages?",
      "answer": "B",
      "parameters": {
        "Entity1": "Pepsi",
        "Entity2": "Coca-Cola",
        "DomainX": "beverages"
      }
    },
    {
      "question": "Are Microsoft and Apple both related to technology?",
      "answer": "B",
      "parameters": {
        "Entity1": "Microsoft",
        "Entity2": "Apple",
        "DomainX": "technology"
      }
    }
  ]
}
```

**User:** Are Ken Kwapis and Jan Švankmajer both filmmakers? Options: A. No B. Yes. Answer: B. Masked Question: Are Person1 and Person2 both ProfessionX? Parameters: 'Person1': 'Christopher Nolan', 'Person2': 'Quentin Tarantino', 'ProfessionX': 'filmmakers'.
**Assistant:**

```
{
  "similar_questions": [
    {
      "question": "Are Christopher Nolan and Quentin Tarantino both filmmakers?",
      "answer": "B",
      "parameters": {
        "Person1": "Christopher Nolan",
        "Person2": "Quentin Tarantino",
        "ProfessionX": "filmmakers"
      }
    },
    {
      "question": "Are J.K. Rowling and Stephen King both authors?",
```

---

```
        "answer": "B",
        "parameters": {
          "Person1": "J.K. Rowling",
          "Person2": "Stephen King",
          "ProfessionX": "authors"
        }
      },
      {
        "question": "Are Beyoncé and Elon Musk both singers?",
        "answer": "A",
        "parameters": {
          "Person1": "Beyoncé",
          "Person2": "Elon Musk",
          "ProfessionX": "singers"
        }
      },
      {
        "question": "Are Henry Ford and Albert Einstein both automobile entrepreneurs?",
        "answer": "A",
        "parameters": {
          "Person1": "Henry Ford",
          "Person2": "Albert Einstein",
          "ProfessionX": "automobile entrepreneurs"
        }
      },
      {
        "question": "Are Tony Hawk and Bob Burnquist both professional skateboarders?",
        "answer": "B",
        "parameters": {
          "Person1": "Tony Hawk",
          "Person2": "Bob Burnquist",
          "ProfessionX": "professional skateboarders"
        }
      }
    ]
  }
```

**User:** Which of these is one similarity of the elements in the first group of the periodic table? Options: A. They all react similarly with water B. They have similar electron configurations. Answer: A. Masked Question: Which of these is one similarity of the elements in Group1 of the periodic table? Parameters: 'Group1': 'first group'.
**Assistant:**

```
{
  "similar_questions": [
    {
      "question": "Which of these is one similarity of the elements in the 18th group of the periodic table?",
      "answer": "B",
      "parameters": {
        "Group1": "18th group"
      }
    },
    {
      "question": "Which of these is one similarity of the elements in the alkali metals of the periodic table?",
      "answer": "B",
      "parameters": {
        "Group1": "alkali metals"
      }
    },
    {
      "question": "Which of these is one similarity of the elements in the noble gases of the periodic table?",
      "answer": "B",
      "parameters": {
        "Group1": "noble gases"
      }
    },
    {
      "question": "Which of these is one similarity of the elements in the halogens of the periodic table?",
      "answer": "A",
      "parameters": {
        "Group1": "halogens"
      }
    },
    {
      "question": "Which of these is one similarity of the elements in the transition metals of the periodic table?",
      "answer": "B",
      "parameters": {
        "Group1": "transition metals"
      }
    }
  ]
```

```
}

User: Which celestial object listed below has the greatest density? Options: A. a planet B. a neutron star. Answer: B. Masked
Question: Which celestial object listed below has the Superlative Property? Parameters: 'Superlative': 'greatest', 'Property':
'density'.
Assistant:

{
  "similar_questions": [
    {
      "question": "Which celestial object listed below has the greatest density?",
      "answer": "A",
      "parameters": {
        "Superlative": "greatest",
        "Property": "density"
      }
    },
    {
      "question": "Which celestial object listed below has the highest temperature?",
      "answer": "A",
      "parameters": {
        "Superlative": "highest",
        "Property": "temperature"
      }
    },
    {
      "question": "Which celestial object listed below has the largest mass?",
      "answer": "B",
      "parameters": {
        "Superlative": "largest",
        "Property": "mass"
      }
    },
    {
      "question": "Which celestial object listed below has the brightest luminosity?",
      "answer": "B",
        "parameters": {
        "Superlative": "brightest",
        "Property": "luminosity"
      }
    },
    {
      "question": "Which celestial object listed below has the longest lifespan?",
      "answer": "B",
      "parameters": {
        "Superlative": "longest",
        "Property": "lifespan"
      }
    }
  ]
}

User: On what would you serve an egg to someone? Options: A. plate B. chicken coop. Answer: A. Masked Question: On what
would you serve a FoodItem to someone? Parameters: 'FoodItem': 'egg'.
Assistant:

{
  "similar_questions": [
    {
      "question": "On what would you serve an egg to someone?",
      "answer": "A",
      "parameters": {
        "FoodItem": "egg"
      }
    },
    {
      "question": "On what would you serve a soup to someone?",
      "answer": "B",
      "parameters": {
        "FoodItem": "soup"
      }
    },
    {
      "question": "On what would you serve a salad to someone?",
      "answer": "B",
      "parameters": {
        "FoodItem": "salad"
      }
    },
    {
```

```
        "question": "On what would you serve a sandwich to someone?",
        "answer": "A",
        "parameters": {
          "FoodItem": "sandwich"
        }
      },
      {
        "question": "On what would you serve a steak to someone?",
        "answer": "B",
        "parameters": {
          "FoodItem": "steak"
        }
      }
    ]
  }
```

**User:** Fighting inflation means being aware of many things, such as what with loans? Options: A. rising interest rates B. loss of money.
Answer: A. Masked Question: Fighting EconomicIssue means being aware of many things, such as what with FinancialAspect?
Parameters: 'EconomicIssue': 'inflation', 'FinancialAspect': 'loans'.
**Assistant:**

```
{
  "similar_questions": [
    {
      "question": "Fighting inflation means being aware of many things, such as what with loans?",
      "answer": "A",
      "parameters": {
        "EconomicIssue": "inflation",
        "FinancialAspect": "loans"
      }
    },
    {
      "question": "Fighting recession means being aware of many things, such as what with unemployment?",
      "answer": "B",
      "parameters": {
        "EconomicIssue": "recession",
        "FinancialAspect": "unemployment"
      }
    },
    {
      "question": "Fighting deflation means being aware of many things, such as what with prices?",
      "answer": "B",
      "parameters": {
        "EconomicIssue": "deflation",
        "FinancialAspect": "prices"
      }
    },
    {
      "question": "Fighting an economic crisis means being aware of many things, such as what with market volatility?",
      "answer": "A",
      "parameters": {
        "EconomicIssue": "economic crisis",
        "FinancialAspect": "market volatility"
      }
    },
    {
      "question": "Fighting a budget deficit means being aware of many things, such as what with government spending?",
      "answer": "B",
      "parameters": {
        "EconomicIssue": "budget deficit",
        "FinancialAspect": "government spending"
      }
    }
  ]
}
```

**User:** NOTE: Make sure the choices and masked question remain the same. Question: {question} Options: {options}. Answer:
{answer}. Masked Question: {masked_question}. Parameters: {entities}.
**Assistant:**

### A.1.4   PROGRAM GENERATION PROMPT FOR SFT

---

**Prompt for *Answer Function Implementation***

**System:** Implement the provided `answer(...)` function exactly as specified by the signature in Python. Return an integer corresponding to the index of the correct option in `Options`. Use the helper function `retrieve(question: str, answer_type: Type) -> Any` only for simple fact-checking. Allowed `answer_type` values: `int`, `float`, `list`, `str`, or `bool`. Do not embed reasoning logic inside `retrieve` calls—just simple fact-check questions. If a single `retrieve` question requires multi-step reasoning, decompose it into multiple simpler `retrieve` calls. Assume `retrieve` is implemented; do not define it. Include every input parameter in your `answer` function. Produce only the `answer` function—no extra code, comments, or imports beyond what's necessary.

**User:** Masked Question: Are Entity1 and Entity2 both related to DomainX? Parameters: `dict_keys(['Entity1', 'Entity2', 'DomainX'])`. Options: `['No', 'Yes']`. Function header: `def answer(Entity1: str, Entity2: str, DomainX: str) -> int.`
**Assistant:**

```python
from typing import Any

def answer(Entity1: str, Entity2: str, DomainX: str) -> int:
    # First check Entity1's relationship
    entity1_related = retrieve(
        f"Is {Entity1} related to {DomainX}?", bool
    )
    # Then check Entity2's relationship
    entity2_related = retrieve(
        f"Is {Entity2} related to {DomainX}?", bool
    )
    # Both must be related for "Yes"
    if entity1_related and entity2_related:
        return 1
    else:
        return 0
```

**User:** Masked Question: Are both founders of CompanyX still involved in the company? Parameters: `dict_keys(['CompanyX'])`. Options: `['No', 'Yes']`. Function header: `def answer(CompanyX: str) -> int.`
**Assistant:**

```python
from typing import Any, List

def answer(CompanyX: str) -> int:
    founders = retrieve(
        f"Who are the founders of {CompanyX}?", list
    )
    still_involved = []
    for founder in founders:
        involved = retrieve(
            f"Is {founder} still involved with {CompanyX}?", bool
        )
        still_involved.append(involved)
    if all(still_involved):
        return 1
    else:
        return 0
```

**User:** Masked Question: Which of these is one similarity of the elements in Group1 of the periodic table? Parameters: `dict_keys(['Group1'])`. Options: `['They all react similarly with water', 'They have similar electron configurations.']`. Function header: `def answer(Group1: str) -> int.`
**Assistant:**

```python
from typing import Any, List

def answer(Group1: str) -> int:
    elements = retrieve(
        f"List the elements in {Group1} of the periodic table", list
    )
    valence_counts = []
    for el in elements:
        count = retrieve(
            f"How many valence electrons does {el} have?", int
        )
        valence_counts.append(count)
    if len(set(valence_counts)) == 1:
        return 1
    water_reactivities = []
    for el in elements:
        reactive = retrieve(
            f"Is {el} reactive with water?", bool
        )
        water_reactivities.append(reactive)
    if len(set(water_reactivities)) == 1:
```

```
        return 0
    return 0
```

**User:** Masked Question: Which celestial object listed below has the Superlative Property? Parameters: `dict_keys(['Superlative', 'Property'])`. Options: `['a planet', 'a neutron star']`. Function header: `def answer(Superlative: str, Property: str) -> int`.
**Assistant:**

```python
from typing import Any, List

def answer(Superlative: str, Property: str) -> int:
    planet_list = retrieve(
        f"Which known planet has the {Superlative} {Property}?", list
    )
    planet_name = planet_list[0]
    planet_value = retrieve(
        f"What is the {Property} of {planet_name} in standard units?", float
    )
    neutron_list = retrieve(
        f"Which known neutron star has the {Superlative} {Property}?", list
    )
    neutron_name = neutron_list[0]
    neutron_value = retrieve(
        f"What is the {Property} of {neutron_name} in the same units?", float
    )
    is_max = retrieve(
        f"Does '{Superlative}' mean selecting the highest {Property}?", bool
    )
    if is_max:
        return 0 if planet_value > neutron_value else 1
    is_min = retrieve(
        f"Does '{Superlative}' mean selecting the lowest {Property}?", bool
    )
    if is_min:
        return 0 if planet_value < neutron_value else 1
    return 0
```

**User:** Masked Question: On what would you serve a FoodItem to someone? Parameters: `dict_keys(['FoodItem'])`. Options: `['plate', 'bowl']`. Function header: `def answer(FoodItem: str) -> int`.
**Assistant:**

```python
from typing import Any

def answer(FoodItem: str) -> int:
    served_on_plate = retrieve(
        f"Is a plate a common item to serve {FoodItem}?", bool
    )
    served_on_bowl = retrieve(
        f"Is a bowl a common item to serve {FoodItem}?", bool
    )
    if served_on_plate:
        return 0
    else:
        return 1
```

**User:** Masked Question: Did HistoricalFigure have ethnicity closer to NativeGroup than ForeignGroup? Parameters: `dict_keys(['HistoricalFigure', 'NativeGroup', 'ForeignGroup'])`. Options: `['No', 'Yes']`. Function header: `def answer(HistoricalFigure: str, NativeGroup: str, ForeignGroup: str) -> int`.
**Assistant:**

```python
from typing import Any

def answer(HistoricalFigure: str, NativeGroup: str, ForeignGroup: str) -> int:
    has_native = retrieve(
        f"Did {HistoricalFigure} have ancestry from {NativeGroup}?", bool
    )
    has_foreign = retrieve(
        f"Did {HistoricalFigure} have ancestry from {ForeignGroup}?", bool
    )
    if has_native and not has_foreign:
        return 1
    if has_foreign and not has_native:
        return 0
    more_native = retrieve(
        f"Did {HistoricalFigure} have a greater proportion of {NativeGroup} ancestry than {ForeignGroup}?", bool
    )
    if more_native:
        return 1
    else:
        return 0
```

```
User: Masked Question: Which reaction did Person have after TimePeriod in a ColdEnvironment? Parame-
ters: dict_keys(['Person','TimePeriod','ColdEnvironment','PhysicalReaction']). Op-
tions: ['chills','calm down']. Function header: def answer(Person: str, TimePeriod: str,
ColdEnvironment: str, PhysicalReaction: str) -> int.
Assistant:

from typing import Any

def answer(Person: str, TimePeriod: str, ColdEnvironment: str, PhysicalReaction: str) -> int:
    had_chills = retrieve(
        f"Did {Person} have chills after {TimePeriod} in a {ColdEnvironment}?", bool
    )
    if had_chills:
        return 0
    had_calm = retrieve(
        f"Did {Person} calm down after {TimePeriod} in a {ColdEnvironment}?", bool
    )
    if had_calm:
        return 1
    return 0

User: Masked Question: Which is a way that Earth loses energy from the EnergySource? Parameters:
dict_keys(['EnergySource']). Options: ['reflection','absorption']. Function header: def
answer(EnergySource: str) -> int.
Assistant:

from typing import Any

def answer(EnergySource: str) -> int:
    source_desc = retrieve(
        f"How does Earth lose energy from {EnergySource}?", str
    )
    is_reflection = retrieve(
        f"Does '{source_desc}' indicate reflection?", bool
    )
    if is_reflection:
        return 0
    is_absorption = retrieve(
        f"Does '{source_desc}' indicate absorption?", bool
    )
    if is_absorption:
        return 1
    return 0

User: NOTE: You must use all input parameters, decompose complex queries into multiple retrieve calls, and ensure at least two
retrieve calls per function. {input}
Assistant:
```

### A.1.5 PROGRAM GENERATION PROMPT FOR RL

**Prompt for *Answer Function Implementation***

**System:** Implement the provided `answer(...)` function exactly as specified by the signature in Python. Return an integer corresponding to the index of the correct option in `Options`. Use the helper function `retrieve(question: str, answer_type: Type) -> Any` only for simple fact-checking. Allowed `answer_type` values: `int`, `float`, `list`, `str`, or `bool`. Do not embed reasoning logic inside `retrieve` calls—just simple fact-check questions. If a single `retrieve` question requires multi-step reasoning, decompose it into multiple simpler `retrieve` calls. Assume `retrieve` is implemented; do not define it. Include every input parameter in your `answer` function. Produce only the `answer` function—no extra code, comments, or imports beyond what's necessary.

```
User: Masked Question: Are both founders of CompanyX still involved in the company? Parameters:
dict_keys(['CompanyX']). Options: ['No', 'Yes']. Function header: def answer(CompanyX: str) -> int.
Assistant:

from typing import Any, List

def answer(CompanyX: str) -> int:
    founders = retrieve(
        f"Who are the founders of {CompanyX}?", list
    )
    still_involved = []
    for founder in founders:
        involved = retrieve(
            f"Is {founder} still involved with {CompanyX}?", bool
        )
        still_involved.append(involved)
    if all(still_involved):
        return 1
    else:
```

```
            return 0
```

**User:** Masked Question: Which celestial object listed below has the Superlative Property? Parameters: `dict_keys(['Superlative', 'Property'])`. Options: `['a planet', 'a neutron star']`. Function header: `def answer(Superlative: str, Property: str) -> int`.

**Assistant:**

```
from typing import Any, List

def answer(Superlative: str, Property: str) -> int:
    planet_list = retrieve(
        f"Which known planet has the {Superlative} {Property}?", list
    )
    planet_name = planet_list[0]
    planet_value = retrieve(
        f"What is the {Property} of {planet_name} in standard units?", float
    )
    neutron_list = retrieve(
        f"Which known neutron star has the {Superlative} {Property}?", list
    )
    neutron_name = neutron_list[0]
    neutron_value = retrieve(
        f"What is the {Property} of {neutron_name} in the same units?", float
    )
    is_max = retrieve(
        f"Does '{Superlative}' mean selecting the highest {Property}?", bool
    )
    if is_max:
        return 0 if planet_value > neutron_value else 1
    is_min = retrieve(
        f"Does '{Superlative}' mean selecting the lowest {Property}?", bool
    )
    if is_min:
        return 0 if planet_value < neutron_value else 1
    return 0
```

**User:** Masked Question: On what would you serve a FoodItem to someone? Parameters: `dict_keys(['FoodItem'])`. Options: `['plate', 'bowl']`. Function header: `def answer(FoodItem: str) -> int`.

**Assistant:**

```
from typing import Any

def answer(FoodItem: str) -> int:
    served_on_plate = retrieve(
        f"Is a plate a common item to serve {FoodItem}?", bool
    )
    served_on_bowl = retrieve(
        f"Is a bowl a common item to serve {FoodItem}?", bool
    )
    if served_on_plate:
        return 0
    else:
        return 1
```

**User:** Masked Question: Did HistoricalFigure have ethnicity closer to NativeGroup than ForeignGroup? Parameters: `dict_keys(['HistoricalFigure', 'NativeGroup', 'ForeignGroup'])`. Options: `['No', 'Yes']`. Function header: `def answer(HistoricalFigure: str, NativeGroup: str, ForeignGroup: str) -> int`.

**Assistant:**

```
from typing import Any

def answer(HistoricalFigure: str, NativeGroup: str, ForeignGroup: str) -> int:
    has_native = retrieve(
        f"Did {HistoricalFigure} have ancestry from {NativeGroup}?", bool
    )
    has_foreign = retrieve(
        f"Did {HistoricalFigure} have ancestry from {ForeignGroup}?", bool
    )
    if has_native and not has_foreign:
        return 1
    if has_foreign and not has_native:
        return 0
    more_native = retrieve(
        f"Did {HistoricalFigure} have a greater proportion of {NativeGroup} ancestry than {ForeignGroup}?", bool
    )
    if more_native:
        return 1
    else:
        return 0
```

**User:** NOTE: You must use all input parameters, decompose complex queries into multiple `retrieve` calls, and ensure at least two `retrieve` calls per function. {input}

### A.1.6 PROGRAM JUDGE & REGENERATION PROMPT (RL)

---

**Prompt for *Program Evaluation and Improvement***

---

**System:** You are given a group of questions derived from the same masked template. Your job is (1) to evaluate the PREVIOUS program and assign a single integer score, and (2) to REGENERATE one improved Python function that solves *all* questions in the group. Think step-by-step. Then output a valid JSON with exactly two keys: {"score": <int>, "program": ""}.

**Tasks (output JSON with "score" and "program"):**
**A) Evaluate the PREVIOUS program and produce ONE integer "score" (1–10) based on three dimensions:**

1. **Factor alignment:** covers important reasoning factors from the Reasoning Path (e.g., alternatives, typical requirements, cultural variability, conservative defaults).

2. **No shortcuts:** no hard-coded mappings/dictionaries; no direct string checks for specific entities; no pattern-matching the literal question text; no label leakage.

3. **Proper decomposition:** breaks the task into simple, orthogonal subquestions via `retrieve()` or equivalent evidence checks that generalize to unseen items (not relying on concrete examples).

*Scoring rubric (guidance):*

- 9–10: strong factor coverage, no shortcuts, clear multi-step decomposition with robust fallbacks.

- 7–8: good factor coverage, minor gaps, mostly clean decomposition.

- 5–6: partial factor coverage and/or weak decomposition.

- 3–4: major gaps; some shortcut-like behavior or brittle logic.

- 1–2: fails most dimensions; relies on prohibited shortcuts or ignores factors.

**B) Generate a NEW program as a single Python function:**

- Signature: `def answer(..)  -> int`. Return `0` or `1`.

- Must use decomposition into general, masked sub-queries (e.g., via `retrieve()`) that do not include concrete items or terms from the questions. Only use placeholders from inputs (e.g., `FoodItem`, `Utensil`) and generic concepts.

- Must not define or rely on any hard-coded mapping/dictionary/list of specific entities; must not read or reference the literal question text.

- Should reflect the Premise and the Inference. Keep the logic self-contained except for a black-box `retrieve(prompt: str, type_hint: type) -> Any`.

- Code should be concise, readable, and deterministic given `retrieve`'s returns; include brief comments.

**OUTPUT FORMAT:**

```
{
  "score": <integer 1-10>,
  "program": "<the improved Python code as a single string>"
}
```

**User:** Masked template

```
- Masked question: "<masked_question>"
- Choices: <choices>
- Options: ["A", "B"]

Questions and results of the previous program

- Q0: "<question_0>" | Gold: <gold_0> | Program: <pred_0>
- Q1: "<question_1>" | Gold: <gold_1> | Program: <pred_1>
...

Reasoning Path (reference facts you should respect)
<reasoning_path_text>

Previous candidate program (to be evaluated and then improved)
<previous_program_code_block>
```

---

### A.2 RL TRAINING

We train our RL models on two H200 GPUs using the VERL framework (Sheng et al., 2025). The retriever runs with temperature 0.7. To characterize memory footprint, we co-locate an Instruct serving endpoint and training on the same two H200s: during backpropagation GPU memory saturates, whereas at inference it uses about half of the capacity. Unless stated otherwise, we use a learning rate of $1 \times 10^{-5}$, a KL-divergence coefficient of 0.001, five rollouts per update, a training batch size of 128, and rollout temperature $= 1$.

## A.3 EXAMPLE

**Prompt**    messages = [ "role": "system", "content": "Let's solve the following question step by step", "role": "user", "content": f"{question}", ]

**Result**    Question: Can you order a Tesla at Dunkin' Donuts?
Answer: Certainly, let́s break this down step by step: 1. **Understanding the Context**: - An Alfa Romeo is a brand of luxury sports cars and sedans. - Starbucks is a global coffee chain known for its beverages, food items, and retail products.2. **Analyzing the Question**: - The question asks if one can "order" an Alfa Romeo at Starbucks. - This implies that the person wants to purchase or acquire an Alfa Romeo from Starbucks.3. **Evaluating Possibilities**: - Starbucks primarily sells beverages, food, and some merchandise related to their brand. - They do not sell vehicles, including cars like Alfa Romeos.4. **Conclusion**: - It is not possible to order an Alfa Romeo at Starbucks because they do not sell automobiles. Therefore, the answer to the question "Can you order an Alfa Romeo at Starbucks?" is no.

Question: Can you order an Alfa Romeo at Starbucks?
Answer: Certainly! Let́s break this down step by step:1. **Identify the Companies**: - Tesla is an electric vehicle manufacturer. - DunkinD́onuts is a coffee and donut chain.2. **Understand Their Business Models**: - Tesla sells vehicles, primarily electric cars, through its own dealerships and online. - DunkinD́onuts sells food and beverages in physical stores and through various digital platforms.3. **Check for Overlap**: - There is no direct business relationship or overlap between Tesla and DunkinD́onuts. They operate in completely different industries.4. **Consider the Question**: - The question asks if you can "order" a Tesla at DunkinD́onuts. This implies purchasing or ordering a product from one company at the location of another.5. **Conclusion**: - You cannot order a Tesla at DunkinD́onuts because Tesla is not a product sold by DunkinD́onuts, and there is no mechanism for ordering a Tesla through a DunkinD́onuts location. Therefore, the answer to the question "Can you order a Tesla at DunkinD́onuts?" is no.

### A.4 THEORY PROOF

*Example ( $N=6$, $K=5$).* Assume conditional independence across variants with per-variant success probabilities

$$\boldsymbol{p} = (0.99, 0.99, 0.99, 0.99, 0.99, 0.20), \quad \boldsymbol{q} = (0.70, 0.70, 0.99, 0.99, 0.99, 0.90).$$

Then

$$R_{\text{normal}}(p) = \tfrac{1}{6}\sum p_i \approx 0.858, \qquad R_{\text{normal}}(q) \approx 0.878 \ (>0.858),$$

but the tail probabilities for $S \geq 5$ satisfy

$$J_5(p) \approx 0.961 \quad \text{versus} \quad J_5(q) \approx 0.855 \ (<0.961).$$

Thus improving per-instance mean accuracy can *reduce* $K$-of-$N$ consistency, while optimizing $R_{\text{coh}}$ directly targets $J_K$. $\square$

**Remark (Monotone lower bound in the mean).** Let $\bar{p} = \frac{1}{N}\sum_i \Pr[Z_i(p){=}1]$. By classical extremal properties of Poisson–binomial sums, $\Pr[S \geq K] \geq \Pr[\text{Binomial}(N, \bar{p}) \geq K]$. Hence increasing $R_{\text{norm}}$ increases a *lower bound* on $J_K$, but—as Proposition 2 shows—does not guarantee improving $J_K$ itself, especially when $K$ is large and success must occur *simultaneously* across many variants.

| Domain | Rejection | Upper-bound Acc |
|---|---|---|
| ARC-Challenge | 72.8 | 96.0 |
| ARC-Easy | 79.7 | 96.0 |
| CSQA | 70.1 | 88.0 |
| StrategyQA | 65.7 | 92.0 |
| HotpotQA | 71.8 | 100.0 |
| SimpleQA | 46.0 | — |

Table 3: Unified sanity checks: Rejection rates(%) during RL training and estimated Upper-bound accuracies(%) after pass@128 sampling with manual program correction.

A.5   SANITY CHECK EXPERIMENT

## A.6 ABLATION STUDY

### A.6.1 TRAINING ON ORIGINAL QUESTIONS ONLY

| Model | ARC-Challenge | ARC-Easy | CSQA | StrategyQA | HotpotQA |
|---|---|---|---|---|---|
| $RL_{Org}$ | $46.6 \pm 4.4$ | $55.4 \pm 4.3$ | $47.6 \pm 4.4$ | $36.8 \pm 4.2$ | $75.4 \pm 3.8$ |
| $RL_{Cohort}$ | $\mathbf{65.8 \pm 4.1}$ | $\mathbf{74.8 \pm 3.8}$ | $\mathbf{73.4 \pm 3.9}$ | $\mathbf{45.8 \pm 4.4}$ | $\mathbf{79.4 \pm 3.5}$ |

Table 4: Lenient accuracy (%) comparison between original (Org) and Cohort RL variant(Cohort). Bold = best

| Model | ARC-Challenge | ARC-Easy | CSQA | StrategyQA | HotpotQA |
|---|---|---|---|---|---|
| $RL_{Org}$ | $34.0 \pm 4.1$ | $42.2 \pm 4.3$ | $37.6 \pm 4.2$ | $20.8 \pm 3.6$ | $52.8 \pm 4.4$ |
| $RL_{Cohort}$ | $\mathbf{56.6 \pm 4.3}$ | $\mathbf{68.0 \pm 4.1}$ | $\mathbf{62.4 \pm 4.2}$ | $\mathbf{31.0 \pm 4.0}$ | $\mathbf{54.2 \pm 4.4}$ |

Table 5: Strict accuracy (%) comparison between original (Org) and Cohort RL variant(Cohort). Bold = best.

### A.6.2 HIGH-QUALITY RETRIEVER & JUDGE

| Model | ARC-Challenge | ARC-Easy | CSQA | StrategyQA | HotpotQA |
|---|---|---|---|---|---|
| 3B-3B | $41.8 \pm 4.3$ | $40.8 \pm 4.3$ | $\underline{42.6 \pm 4.3}$ | $\mathbf{44.2 \pm 4.3}$ | $15.0 \pm 3.1$ |
| 3B-7B | $\underline{42.0 \pm 4.3}$ | $\underline{41.0 \pm 4.3}$ | $42.2 \pm 4.3$ | $\underline{44.0 \pm 4.3}$ | $\underline{18.6 \pm 3.4}$ |
| 7B-7B | $\mathbf{56.8 \pm 4.3}$ | $\mathbf{65.6 \pm 4.1}$ | $\mathbf{65.6 \pm 4.1}$ | $39.2 \pm 4.3$ | $\mathbf{23.6 \pm 4.2}$ |

Table 6: Lenient accuracy (%) across reasoning datasets with Qwen2.5-7B-Instruct serving as both retriever and judge for the 3B model. "3B-3B" refers to models trained and evaluated with the 3B retriever and judge; "3B-7B" refers to models trained with the 3B retriever/judge and evaluated with the 7B retriever; "7B-7B" refers to models trained and evaluated with the 7B retriever. **Bold** = best, underline = second best.

| Model | ARC-Challenge | ARC-Easy | CSQA | StrategyQA | HotpotQA |
|---|---|---|---|---|---|
| 3B-3B | $\underline{35.4 \pm 4.2}$ | $37.0 \pm 4.2$ | $\underline{36.2 \pm 4.2}$ | $\mathbf{32.6 \pm 4.1}$ | $7.6 \pm 2.3$ |
| 3B-7B | $\underline{35.4 \pm 4.2}$ | $37.4 \pm 4.2$ | $35.8 \pm 4.2$ | $\mathbf{32.6 \pm 4.1}$ | $\underline{11.4 \pm 2.8}$ |
| 7B-7B | $\mathbf{47.8 \pm 4.4}$ | $\mathbf{56.4 \pm 4.3}$ | $\mathbf{56.8 \pm 4.3}$ | $\underline{25.8 \pm 3.8}$ | $\mathbf{17.4 \pm 4.3}$ |

Table 7: Strict accuracy (%) across reasoning datasets with Qwen2.5-7B-Instruct serving as both the retriever and judge for the 3B model. Naming and notes follow Table 6. **Bold** = best, underline = second best.

### A.6.3 OUT OF DOMAIN DATA ACCURACY

| Model | OpenBookQA | PubMedQA | MMLU |
|---|---|---|---|
| Vanilla | $19.0 \pm 3.1$ | $30.0 \pm 4.0$ | $29.8 \pm 4.0$ |
| $SFT_{DM-5k}$ | $57.6 \pm 4.3$ | $10.8 \pm 2.7$ | $40.0 \pm 4.3$ |
| $RL_{Normal}$ (Exec+Crit) | $\mathbf{79.6 \pm 3.5}$ | $\underline{41.2 \pm 4.3}$ | $\underline{62.2 \pm 4.2}$ |
| $RL_{Cohort}$ (Exec+Crit) | $\underline{79.4 \pm 3.6}$ | $\mathbf{50.0 \pm 4.4}$ | $\mathbf{62.4 \pm 4.2}$ |

Table 8: Out-of-distribution (OOD) single-question accuracy (%) on OpenBookQA, PubMedQA, and MMLU. **Bold** = best, underline = second best.

### A.7  HUMAN ANALYSIS

| *Accuracy (%)* | | *Win Rate (%)* | | |
|---|---|---|---|---|
| Answer Label | Abstraction Match | SFT | $RL_{Normal}$ | $RL_{Cohort}$ |
| 92.0 | 96.7 | 23.0 | 30.0 | 47.0 |

Table 9: Human evaluation of generated questions and reasoning paths, reporting both accuracy metrics and win rates for different models.

| Method | Ambiguity / Annotation | Control-Flow / Syntax | Retrieve |
|---|---|---|---|
| $RL_{Cohort}$ | 38 | 54 | 8 |
| $SFT_{DM-5k}$ | 38 | 20 | 42 |

Table 10: Failure type distribution(%) on failure questions. RL reduces Retrieve errors while shifting residual errors to Control-Flow/Syntax.

### A.8  LLM USAGE

During manuscript preparation, we used a general–purpose large language model (OpenAI ChatGPT) only for language polishing and structural editing. Concretely, we asked the model to: improve fluency and clarity at the sentence/paragraph level (grammar, wording, concision), suggest local reordering for better flow (e.g., merging redundant sentences, moving a definition earlier) and help finding related works. All edits were treated as suggestions and were reviewed, accepted, or rewritten by the authors.

