# OpenReview forum: "CC-Learn: Cohort-Based Consistency Learning"
_ICLR.cc/2026/Conference — Submitted to ICLR 2026_

### Official Review · Reviewer_F4UB · 2025-10-30

**Soundness:** 3
**Presentation:** 3
**Contribution:** 3
**Rating:** 4
**Confidence:** 4

**Summary:**

This paper introduces CC-LEARN, an RL framework designed to improve reasoning consistency in LLMs. It trains models on cohorts of similar questions that share the same reasoning path but differ in factual details, forcing one executable program to solve all variants. Using GRPO, the model optimizes a composite reward that combines execution-based and critique-based signals, including accuracy, retrieval efficiency, and structural alignment from a frozen judge. The approach yields significant gains in reasoning stability and generalization across multiple QA benchmarks.

**Strengths:**

Strengths

* Well written and easy to follow – The paper is clearly structured, making the methodology and training pipeline easy to understand.

* Good motivation – The authors clearly articulate the importance of improving reasoning consistency in LLMs and provide strong justification for their approach.

* Interesting approach – The use of executable program generation to enforce shared reasoning paths across cohorts is a fresh and creative idea for me, and I think this adds novelty to this paper. Also, although the reward function contains 6 different terms which can hinder the model to be learn properly due to the complex signal, each term seems to be reasonable.

* Reliable experimental results – The inclusion of standard deviation values in the results supports the reliability and robustness of the reported performance.

**Weaknesses:**

Weaknesses and Questions

- Complex Reward Design – The reward model contains too many distinct components (execution-based and critique-based signals), making the overall optimization signal quite complex. It is unclear whether each term meaningfully contributes to the final behavior or if some components might introduce noise during training.

- Heuristic Reward Scaling – The scaling coefficients of individual rewards (e.g., 0.2 per correct answer, 0.06 for decomposition score) appear to be set heuristically without principled tuning or sensitivity analysis. It would be helpful to understand how robust the method is to different reward weightings.

- Cohort Example Ambiguity (Appendix A.3) – In the provided examples, the answers across two questions seem swapped (the answer for the first question is written under the second question and vice versa :)). Anyway, the main point is that it is not entirely clear whether the generated program solves all cohort variants through the same reasoning steps as claimed. For me, it seems the model solves two cohorts in different logic. Clarification is needed on how consistency manifests across variants.

- StrategyQA-Specific Improvement – In Table results, other baselines perform poorly on StrategyQA, whereas the RL_{cohort}(Exec + Crit) variant shows a large jump. What specific aspect of StrategyQA or the reward composition enables this sharp improvement? A deeper analysis would help understand when the proposed RL is more helpful for training.

- Limited Analysis of Failure Cases – While the paper provides some qualitative examples, it lacks a systematic discussion of where CC-LEARN fails or produces inconsistent reasoning. Understanding failure modes could clarify the limitations of the method and guide future refinements.

**Questions:**

See the above section. If the authors provide convincing explanations to my questions, I would be willing to raise my score, as the overall idea and motivation are good.

---

> ### Author Response · Authors · 2025-11-19
> **Rebuttal [1/2]**
>
> We thank the reviewer for the valuable feedback and provide responses to each point below.
>
> **Weakness 1:**
>
> Consistency in LLM reasoning is hard to achieve in the first place, as many works have shown ([1], [2], [3], [4]). As a result, we employ our proposed reward design, each component has a clear motivation and role, and they work together to make consistent learning feasible. The high-level structure contains two clearly defined major parts: cohort accuracy to encourage consistency, and judge-based signals to facilitate learning. All other rewards serve very specific purposes to make sure such learning can happen. For the verifiable execution rewards, we found that once we removed R_ret (decomposition reward), it drives the policy model to ask a simple paraphrase of the input question. When we remove another reward, R_rej (rejection penalty), the policy model will try to re-ask the same input question. Neither of the generations is what we expected; what we want is a proper decomposition of the input question. For the judge-based rewards, we aim to utilize these rewards to provide additional signals to the policy model when the generations fail but still offer some benefits that the model can learn from (for example, a better control flow or a better coverage of relevant factors). Empirically, as shown in the tables, the model benefits from adding critique rewards, indicating that these rewards provide useful signals during training.
>
>
>
> **Weakness 2:**
>
> First, GRPO work on the relative scales of rewards within the group, so our heuristics mostly guarantee a relative comparison between what we view as "good" and "bad" rollouts. As a result, the specific tuning of individual reward weightings will lead to minimal gains, so we did not spend efforts on that. In our reward design, the accuracy reward is weighted 2× any other single reward; all other rewards share the same magnitude cap (e.g., 0.6). We chose this reward design that prioritizes accuracy over any other single reward, including both verifiable execution rewards and critique rewards. We want non-accuracy rewards to shape the learning while focusing on the cohort accuracy. Our goal here is to show how to improve the consistency with our cohort design rather than identifying the best hyperparameters specifically with our training data; future work is likely not going to apply in the exact same setting.
>
>
> **Weakness 3:**
>
> Thanks for pointing that out! We will fix it in the camera-ready version. This example consists of two responses from the end-to-end model and illustrates why do we need to enforce consistency in LLM. The masked question is “Masked Question: Can you order a ProductX at RestaurantY?” and the two questions are “Can you order an Alfa Romeo at Starbucks?” and “Can you order a Tesla at Dunkin’ Donuts?”. For the first question, the model may answer No by invoking a product-availability path (coffee shops do not sell cars / not on the menu). However, for the second question, the same model may again answer No but justify it via a different partnership path (no business relationship between Tesla and  Dunkin’ Donuts ⇒ not available). From a human’s perspective, none of the reasoning paths is complete and covers all the factors. However, with the same program, we can enforce the model to solve similar questions with the same reasoning path, which makes it more consistent across different variants.
>
>
>
> [1] Mirzadeh, I., Alizadeh, K., Shahrokhi, H., Tuzel, O., Bengio, S., & Farajtabar, M. (2024). GSM-Symbolic: Understanding the limitations of mathematical reasoning in large language models (arXiv preprint arXiv:2410.05229 v2). https://doi.org/10.48550/arXiv.2410.05229
>
>
> [2] Omar, M., Soffer, S., Agbareia, R., et al. (2025). Sociodemographic biases in medical decision making by large language models (Nature Medicine, 31, 1873–1881). https://doi.org/10.1038/s41591-025-03626-6
>
>
> [3] Uceda-Sosa, R., Natesan Ramamurthy, K., Chang, M., & Singh, M. (2024). Reasoning about concepts with LLMs: Inconsistencies abound (arXiv preprint arXiv:2405.20163 v1). https://doi.org/10.48550/arXiv.2405.20163
>
>
> [4] Xie, Z., Guo, J., Yu, T., & Li, S. (2024). Calibrating reasoning in language models with internal consistency (NeurIPS 2024; arXiv preprint arXiv:2405.18711 v2). https://doi.org/10.48550/arXiv.2405.18711

---

> ### Author Response · Authors · 2025-11-19
> **Rebuttal [2/2]**
>
> **Weakness 4:**
>
>
> The main reason is that stqa is more aligned with our goal. In fact, stqa consists of implicitly compositional questions(Just like the example we show in W3). Those questions need to be solved via proper decompositions. Our objective is to encourage the model to generate a correct reasoning path via proper decomposition of the original question. Thus, stqa is well aligned with our goal. Another reason is that both models have a low starting point on the stqa dataset, which makes the improvement so large.
>
> **Weakness 5:**
>
> As we mentioned in the ablation study (Human Study: Failure Case Analysis), the majority of failure cases of CC-Learn arise from control flow/syntax issues. Control flow means the program’s logic is wrong (e.g, wrong and/or aggregation, branching/looping, etc), and it accounts for 34% of all the failure questions. Syntax refers to the model using incorrect syntax, which accounts for 20% of all failure questions. We provide a preliminary analysis in the human study and Appendix A.7 and will elaborate this section with more details in the camera-ready version.

---

### Official Review · Reviewer_Xbz4 · 2025-10-31

**Soundness:** 3
**Presentation:** 2
**Contribution:** 2
**Rating:** 6
**Confidence:** 2

**Summary:**

The paper introduces CC-LEARN, a cohort-based reinforcement learning framework aimed at improving consistency in reasoning for large language models. It abstracts similar questions into masked templates, generates reusable executable reasoning programs, and employs atomic retrieval operations with rejection filtering. Experimental results demonstrate substantial accuracy improvements and enhanced consistency across diverse benchmarks compared to baseline methods.

**Strengths:**

- The paper introduces a framework that groups semantically similar questions into cohorts and trains the model to produce a single executable reasoning program shared across them. This design directly targets consistency by forcing uniform reasoning across paraphrased inputs, reducing random output variance and improving stability in reasoning behavior.

- Training is guided by a composite reward that combines execution accuracy, retrieval efficiency, and structural critiques from a frozen judge model. This dual-signal design rewards both correct outcomes and logically complete, well-aligned reasoning, effectively encouraging the model to generate reliable and factor-aware programs.

- Using cohort-based K-of-N evaluation, the paper demonstrates that CC-LEARN maintains consistent reasoning across rephrased or structurally similar questions.

**Weaknesses:**

- K-of-N scoring rewards group success but may mask problematic individual cases. Absent calibrated confidence or human-in-the-loop triggers, ambiguous items can be mishandled. That limits suitability in high-stakes settings.

- Similar-question cohorts and SFT programs are synthesized by frontier LLMs, then only a small sample is human-checked, which risks template artifacts or label bias leaking into both train and test.

- Figure 2 contains overlapping text.

**Questions:**

Please check the weaknesses.

---

> ### Author Response · Authors · 2025-11-19
> **Rebuttal**
>
> We thank the reviewer for the valuable feedback and provide responses to each point below.
>
> **Weakness 1:**
>
> The risk of models learning to mask problematic individual cases is very low under our K-of-N setting. This is because the N and K we use are relatively big (6 and 4) in training, and for an individual training question with a relatively limited scope, being able to answer 4 questions correctly in the cohort is a strong indicator that the model has learned a generalizable solution for the specific question. We will conduct a manual analysis to demonstrate this in the final version. An empirical reason that we employ a K of N training setup is because of the label noise in the generated similar questions (8%, which is on par with many benchmarks today). Because of these noises, an optimal model cannot guarantee to solve all 6 out of 6. Imposing perfect consistency may cause over-penalization, which we do not want. Another reason that we believe a k-of-n setup is nicely designed to encourage a consistent solution is as we have shown in Propositions 1 and 2, our K-of-N training setting aligns more closely with our consistency evaluation than the standard accuracy calculation (Appendix A.4). As a result, empirically, as Table 8 shows, even with a 4-out-of-6 training setting, models can generalize to higher consistency requirements during evaluation (strict setting uses 5-out-of-6), showcasing the robustness of our reward design and further mitigate the concern of the reviewer.
>
> **Weakness 2:**
>
>
> Under our data generation pipeline, there is nearly no risk of template artifacts or label bias leakage. For the template artifacts, during data generation, our abstractions are not fixed templates reused across questions. Instead, they are abstractions of individual questions drawn from a large source pool (including chemistry, physics, literature, math, art, etc). Each abstraction has different entities, parameters, and so on. As a result, the diversity of the abstraction is high, with minimal to no risk in artifacts just from the templates. For the label leakage, we cross-validate the labels against three 70B models and use the agreement to confirm correctness, which minimizes the risk of leakage from any single model’s biases.
>
>
>
> **Weakness 3:**
>
> Thanks for flagging. We will fix the overlapping text in Figure 2 in the camera-ready version!

---

### Official Review · Reviewer_Qa3J · 2025-11-01

**Soundness:** 2
**Presentation:** 3
**Contribution:** 3
**Rating:** 4
**Confidence:** 4

**Summary:**

This paper proposes an approach to ensure that LLMs produce robust reasoning by training on a cohort of similar examples instead of a single example in which the model could be rewarded for arriving at the right answer through the wrong reasoning steps. The approach consists of using a composite objective function that combines execution based signals like accuracy and retrieval usage that are easily measurable, and critique based signals like factor-complete decomposition that require a frozen LLM judge. Using GRPO to train in this cohort-based way, the approach shows promising results on in-distribution and out-of-distribution benchmarks.

**Strengths:**

- The paper addresses an important issue in LLM reasoning, where the model can be potentially rewarded for providing an incorrect reasoning because the final answer it outputs passes problem-specific tests.
- The method proposed to address this issue is innovative and builds on prior work in deep learning literature in the context of LLM reasoning: ensuring that the solution produced generalises across a cohort of samples instead of a single sample, which makes it much harder for the model to chance upon a solution with reasoning (in this case a piece of code) satisfying the entire cohort.
- The use of code as the reasoning trace is an interesting and novel choice because it forces the reasoning to be fully objective, which makes it executable and also interpretable.

**Weaknesses:**

- For most results in table 1, the 7B model seems to show similar performance for RL-normal and RL-cohort with execution-based and critique-based rewards (within confidence interval). Since the gap is not statistically significant, it seems to indicate that the gains compared to the rest of the baselines for the 7B model are due to the reward function design, not cohort-based consistency enforcement.
- Most of the statistically significant gains in Table 1 seem to be for the 3B model, so studying the effect of the proposed method in the context of the model size might give more insights into the actual behaviour of the method.
- A key weakness of this paper is the lack of a discussion on what it means to impose human-interpretable consistency to constrain the behaviour of LLMs and the reasoning patterns they produce. The idea of enforcing consistency in the reasoning trace through similar examples semantically and expressing them as code is appealing, but LLMs have shown emergent properties as well, which may be impacted by this type of cohort-based learning.
- Choosing definitions for lenient accuracy to mean that 4 out of 6 tests passed and strict accuracy to mean 5 out of 6 tests passed seems very cherry-picked and is not backed up by any convincing reason. It might be more meaningful in terms of results to observe the entire spectrum of test pass rate.
- While there is discussion on prior work in LLM reasoning, this paper would benefit from a subsection on similar approaches to enforce consistency in pre-LLM deep learning literature.

**Questions:**

- The choice of strict accuracy as being able to pass 5 out of 6 tests seems quite arbitrary. Given that strict accuracy usually means that all tests pass, would the authors clarify why this particular decision was taken? It would be helpful to see results with strict accuracy meaning all tests passed.
- How do the retrieve function usage reward and the rejection penalty interact with each other, since one is rewarding correct decomposition and the other is penalising very simple queries? Is there a reason for why the coefficients for these are 0.6 and -0.1 respectively? More broadly, can the authors discuss the rationale behind the coefficients for the reward function and whether or not there were ablations conducted on different values of these coefficients?
- One interesting insight from the paper is that a larger retriever at eval time is always better than a smaller one. Have the authors tried a 7B-3B version, where the eval retriever is smaller than the one used in training?

---

> ### Author Response · Authors · 2025-11-19
> **Rebuttal [1/2]**
>
> We thank the reviewer for the valuable feedback and provide responses to each point below.
>
> **Weakness 1 & 2:**
>
> While the improvement of 7B models under the lenient metric is limited, we observe more significant improvements under the strict metric. As shown in Table 1 and Table 2, the gap between RL-normal and RL-cohort increases from 2% to 4%. This can be explained by the strict metric being more sensitive to consistency within a cohort, indicating that our cohort design helps RL training and encourages the policy model to generate more consistent programs. 3B models, on the other hand, have a worse consistency out of the box, so they demonstrate larger gains in the more lenient evaluation setting. As this trend shows, we could have more similar questions to form a cohort and evaluate our model under a stricter metric. This gap will likely increase further, which highlights the benefits of the cohort design in improving consistency. Additionally, as both propositions 1 and 2 demonstrate, RL-cohort aligned better, theoretically, with the consistency metric than RL-normal.
>
> **Weakness 3:**
>
> As many works have pointed out ([1], [2], [3], [4]), models (including SOTA ones) may employ a totally different reasoning path on question groups where people deem to be identical in reasoning and logic (e.g., simple math questions with different numbers, medical case inputs with different names). In many high-stakes settings, such behaviors are not tolerable as consistency is more important than accuracy. For example, medical experts design evaluation suites to audit whether an AI model can be applied for certain medical needs, such as diagnosis aid. They design the evaluation questions based on the assumption that models will consistently perform across question groups that humans deem to be similar, so human experts are confident that when models do well on the evaluation questions, they will also perform well on newer cases that are similar. If models cannot deliver this consistency guarantee, they cannot be trustworthy in these types of deployments under such auditing (which the most common way to audit tools in the medical field); this is precisely the problem that we are trying to solve, and why imposing human-interpretable consistency is very important as a research direction, even though models demonstrate some level of consistency already.
>
>
> **Weakness 4 & Question 1:**
>
> We employ 5 out of 6 instead of all 6 for two reasons. First, as Table 9 shows, the evaluation questions' gold labels are not perfect (92%, which is on par with many benchmarks today), and the error rate (8%) is far lower than 1/6. This means that a perfectly consistent model can achieve 5/6, but may not achieve 6/6 due to a small amount of label noise. Second, 5 out of 6 is already challenging enough, with models demonstrating 60's overall accuracy. Future works can impose better label accuracy and extend to 6/6 if needed, with minimal effort.
>
> **Weakness 5:**
>
> Thanks for your suggestion. We will add a subsection on pre-LLM consistency methods in the camera-ready version. While no pre-LLM work we know of directly inspires or is relevant to our paper, we fully agree that future readers may benefit from such discussions.
>
>
> References:
>
> [1] Mirzadeh, I., Alizadeh, K., Shahrokhi, H., Tuzel, O., Bengio, S., & Farajtabar, M. (2024). GSM-Symbolic: Understanding the limitations of mathematical reasoning in large language models (arXiv preprint arXiv:2410.05229 v2). https://doi.org/10.48550/arXiv.2410.05229
>
>
> [2] Omar, M., Soffer, S., Agbareia, R., et al. (2025). Sociodemographic biases in medical decision making by large language models (Nature Medicine, 31, 1873–1881). https://doi.org/10.1038/s41591-025-03626-6
>
>
> [3] Uceda-Sosa, R., Natesan Ramamurthy, K., Chang, M., & Singh, M. (2024). Reasoning about concepts with LLMs: Inconsistencies abound (arXiv preprint arXiv:2405.20163 v1). https://doi.org/10.48550/arXiv.2405.20163
>
>
> [4] Xie, Z., Guo, J., Yu, T., & Li, S. (2024). Calibrating reasoning in language models with internal consistency (NeurIPS 2024; arXiv preprint arXiv:2405.18711 v2). https://doi.org/10.48550/arXiv.2405.18711

---

> ### Author Response · Authors · 2025-11-19
> **Rebuttal [2/2]**
>
> **Question 2:**
>
> In our reward design, we weight the accuracy reward 2× any other single reward to ensure it dominates the objective. We cap the magnitude of all other rewards at 0.6. For the penalty reward, it is not -0.1. Instead, it is -0.1*number of failed rejections across cohorts (which ranges from [0, 6]). Thus, it also ranges from [-0.6, 0], with a magnitude cap of 0.6. Furthermore, because GPRO computes relative advantages, training does not depend on absolute scale; a sensitive study on the absolute value is out of our scope. Our goal here is to show how to improve consistency in our cohort design rather than identifying the best hyperparameters specifically for our training data; future work may adopt different reward scales and settings.
>
> **Question 3:**
>
>
> For RL training, our rewards primarily rely on symbolically verifiable rewards. Thus, even if a weaker judge generates some noise, it cannot outweigh the verifiable rewards. Empirically, as Tables 1 and 2 show, when we remove the critique rewards entirely, the model can still surpass the SFT and RL baselines (10-15% improvement). This shows that our method can improve only on verifiable rewards, and the critique rewards can only shape learning, not dominate it. Additionally, our “High-Quality Retriever & Judge” study reveals that stronger judges generally contribute to improvements, but these benefits persist even with a weaker judge for the retrieval function. This indicates that the framework is robust to the judge's strength. For the evaluation, we also experimented on 7B-3B version. In summary, the 7B-3B version performs better than the 3B-3B version but worse than the 7B-7B version. We will add detailed results in the camera-ready version.

---

### Official Review · Reviewer_GjQF · 2025-11-01

**Soundness:** 2
**Presentation:** 2
**Contribution:** 3
**Rating:** 4
**Confidence:** 2

**Summary:**

The paper proposes CC-Learn, a new reinforcement learning framework to fix a key weakness in large language models (LLMs): reasoning inconsistency. LLMs often answer one question correctly but fail on a rephrased or similar version because they use different, fragile reasoning steps. To address this, CC-Learn trains an LLM to produce one Python program that must solve a whole group (or “cohort”) of semantically similar questions. CC-Learn achieves 10–20+ point gains in both accuracy and reasoning consistency over strong supervised baselines on several benchmarks (e.g., ARC, CSQA, StrategyQA, MMLU, PubMedQA).

**Strengths:**

- The approach delivers large absolute improvements (10–20+ points) compared to strong baselines, consistent across various datasets, model scales (3B and 7B), and evaluation settings (Lenient and Strict).

- The model also performs strongly on out-of-domain benchmarks, indicating that the reasoning strategies learned during training generalize well to unseen tasks and domains.

**Weaknesses:**

1. **High reliance on subjective critique rewards:**
   CC-LEARN’s performance depends heavily on two critique-based rewards, ( $R_{fc}$ and $R_{sa}$ ), both provided by a “Judge” model. $R_{fc}$ is a qualitative score (1–10) reflecting how well the Judge thinks the program covers key factors—an inherently subjective judgment. $R_{sa}$ measures structural similarity to an “improved” program $p^{+}$ that the Judge itself creates, meaning the model is rewarded for resembling the Judge’s output rather than being truly correct. As a result, the policy’s performance is effectively limited by the Judge’s own reasoning and programming ability.

2. **Narrow applicability and rigid design:**
   The method is well-suited for tasks that can be neatly expressed as Python programs using atomic `retrieve` calls (e.g., QA or commonsense reasoning), but its generalization to open-ended or creative tasks remains unclear. Moreover, the strict enforcement of “atomic” retrievals—defined through a few-shot prompt—may reduce flexibility, causing the model to produce unnecessarily long or error-prone code when a slightly more complex retrieval would suffice.

**Questions:**

1. Since the policy's quality is guided by the Judge's critique ($R_{sa}$) and ablations show stronger Judges yield stronger policies, what happens if the Judge model is weaker than the Policy model? Is the framework capable of "self-improvement" where the policy can surpass the quality of its Judge?

---

> ### Author Response · Authors · 2025-11-19
> **Rebuttal**
>
> We thank the reviewer for the valuable feedback and provide responses to each point below.
>
> **Weakness 1:**
>
> Our reward design has both symbolically verifiable rewards (i.e., cohort accuracy, retrieval usage, rejection penalty) and model-based critique rewards. Our method can fully work and demonstrate effectiveness without model-based critique rewards. As shown in Tables 1 and 2, with only verifiable rewards, our method still outperforms the SFT and RL baselines by 10-15%. Thus, the critique rewards are designed as a “nice-to-have” augmentation that provides partial signals to facilitate model learning in extremely challenging cases where it is hard for models to generate good programs out of the box; it also offers insights to future works. In fact, our RL design guarantees that the critique rewards will not outweigh verifiable ones, so that the expected noise and subjectivity from the critique models will not undermine the robustness of our model learning or diminish the contribution of our proposed framework.
>
> **Weakness 2:**
>
> While we study a predefined set of benchmarks, our pipeline design is not narrow in applicability, and it is not rigid. First, the program itself can be easily applied in other domains with minimal engineering effort. Both [1] and [2] demonstrate that calling different APIs within a program can make it applicable to diverse scenarios. For example, API calls could include database lookups, calculator operations, code execution, or retrieval of guidelines/KBs. Second, our retrieval function itself is not a rigid design; it does not include too many constraints. Thus, it can be implemented/adapted to different needs and scenarios. Additionally, in high-stakes scenarios such as medical, legal, and financial fields, where consistency is more important than accuracy, our pipeline is well-suited even without adaptation. With only minimal engineering effort, they can be applicable for high-stakes scenarios as well.
>
> **Question 1:**
>
> As we mentioned earlier, our rewards primarily rely on symbolically verifiable rewards. Thus, even if a weaker judge generates some noise signals, it cannot outweigh the verifiable rewards. Empirically, as Tables 1 and 2 show, when we remove the critique rewards entirely, the model can still surpass the SFT and RL baselines (10-15% improvement). This shows that our method can improve only on verifiable rewards, and the critique rewards can only shape learning, not dominate it. Additionally, our “High-Quality Retriever & Judge” study reveals that stronger judges generally contribute to improvements, but these benefits persist even with a weaker judge for the retrieval function. This indicates that the framework is robust to the judge's strength.
>
>
>
> Reference:
>
> [1] Patil, S. G., Zhang, T., Wang, X., & Gonzalez, J. E. (2024). Gorilla: Large language model connected with massive APIs (NeurIPS 2024; arXiv preprint arXiv:2305.15334). https://doi.org/10.48550/arXiv.2305.15334
>
> [2] Wang, G., Xie, Y., Jiang, Y., Mandlekar, A., Xiao, C., Zhu, Y., Fan, L., & Anandkumar, A. (2023). Voyager: An open-ended embodied agent with large language models (arXiv preprint arXiv:2305.16291). https://doi.org/10.48550/arXiv.2305.16291

---

### Meta-Review · Area_Chair_KiUU · 2026-01-07

**Summary:**

The paper proposes to the fix the problem in RL where a model gives a correct answer for one question, but makes a mistake in a similar re-worded question. The key intuition is that instead of using fragile reasoning steps per related problem independently, the model trains an LLM to produce one Python program that solves a cohort of semantically related questions. Using verifiable reward and LLM-as-a-judge based consistency reward signals, the model is trained using GRPO. It shows strong results in both in-distribution and out-of-distribution benchmarks.

The paper received a 4, 4, 4, and 6. All reviewers praise the motivation of the problem, novelty of the solution and strength of experimental results. Most of the criticisms are in (1) details of methodology, (2) critique on the specific reward structure and associated hyperparameters, (3) evaluation metrics, (4) some clarity issues, and (5) questions on scaling of insights from this paper.

The authors give good responses, and they seem mostly convincing to me. The key questions that remain in my mind are: (A) should there have been a stronger hyperparameter tuning to understand the relative strengths of two rewards. The specific choices feel arbitrary. (B) There are five reward components. In an ideal world, one would have removed each reward component in an ablation study to show the value of each component. Otherwise the reward does feel overly complex (as one reviewer indicated). I am not convinced by the author response, and only 1 ablation is shown in the paper. (C) There is concern on whether these insights will extend to a larger model. Now, I know that this is a slightly unfair question, since most researchers can only train up to 7B parameter models and for most research, one does not know whether the improvements will stay or not when models get scaled. In this case, however, there appears to be a significant difference in the amount (and nature) of benefit in 3B and 7B parameter models... bringing this issue into question.

In light of the observations above, and in light of the fact that none of the authors are overly excited by the paper (no one gave a 7 or higher early on), I suggest that the paper be sent for a further revision (with expectations of deeper analysis) and resubmission. That said I will NOT be unhappy if the paper gets accepted.

**Reviewer Concerns:**

see above for the concerns that are still open in my mind.

**Reviewer Scores:**

I expect two 4s to move up to 5 or 6. But, I do not expect one 4 (F4UB) to move up. They ask a good question about complexity of rewards, the response for which I dont feel very satisfied with.

---

### Decision · Program_Chairs · 2026-01-26

Reject